# MMTok: Multimodal Coverage Maximization for Efficient Inference of VLMs

**Sixun Dong**[1][†]    **Juhua Hu**[2]    **Mian Zhang**[3][†]    **Ming Yin**[4][†]    **Yanjie Fu**[1]    **Qi Qian**[5][✉]

[1]Arizona State University    [2]University of Washington    [3]University of Texas at Dallas
[4]Duke University    [5]Zoom Communications

{sixundong.ai,qianq.mail}@gmail.com,juhuah@uw.edu,yanjie.fu@asu.edu

## Abstract

Vision-Language Models (VLMs) demonstrate impressive performance in understanding visual content with language instruction by converting visual inputs to vision tokens. However, redundancy in vision tokens results in the degraded inference efficiency of VLMs. While many algorithms have been proposed to reduce the number of vision tokens, most of them apply only unimodal information (i.e., vision/text) for pruning and ignore the inherent multimodal property of vision-language tasks. Moreover, it lacks a generic criterion that can be applied to different modalities. To mitigate this limitation, in this work, we propose to leverage both vision and text tokens to select informative vision tokens by the coverage criterion. We first formulate the subset selection problem as a maximum coverage problem. Afterwards, a subset of vision tokens is optimized to cover the text tokens and the original set of vision tokens, simultaneously. The proposed method MMTok is extensively evaluated on benchmark datasets with different VLMs. The comparison illustrates that vision and text information are complementary, and combining multimodal information can surpass the unimodal baseline with a clear margin. Moreover, under the maximum coverage criterion on the POPE dataset, our method achieves a 1.87× speedup while maintaining 98.7% of the original performance on LLaVA-NeXT-13B. Finally, with only four vision tokens, 87.7% of the original performance is still preserved on LLaVA-1.5-7B. These results highlight the effectiveness of coverage in token selection. The code is available at https://github.com/Ironieser/mmtok

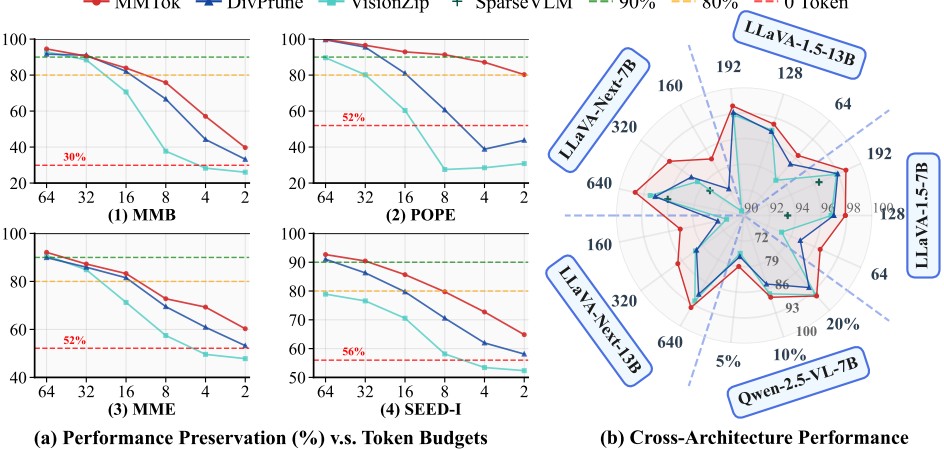

Figure 1: MMTok demonstrates better performance across multiple benchmarks.

---

† Work done during internship at Zoom Communications.
✉ Corresponding author.

# 1 INTRODUCTION

By converting the visual input to vision tokens, Vision-Language Models (VLMs) can leverage powerful Large Language Models (LLMs) to understand visual content as text (Liu et al., 2024b; Li et al., 2024b; Team et al., 2023). Unlike discrete text tokens, where the information is highly compressed, current vision encoders extract vision tokens directly from the original input patches, which are redundant according to previous studies (Bolya et al., 2022; He et al., 2022) and their count can far exceed that of text tokens. For example, given "Describe the image" with less than 10 text tokens, 2,880 vision tokens can be obtained from a single image in LLaVA-NeXT (Liu et al., 2024a).

Since LLMs are built on self-attention layers (Vaswani et al., 2017) that have a quadratic computational cost with respect to the total number of tokens, the large volume of vision tokens can significantly challenge the inference efficiency of VLMs. To accelerate inference, many works (Shang et al., 2024; Yang et al., 2025a; Zhang et al., 2024) have been proposed to sample a subset of vision tokens for inference without compromising performance. While some work adopts an additional training process (Yang et al., 2025a) to enable vision token selection, in this work, we will focus on the training-free paradigm to reduce optimization efforts. Our experiments also confirm that the proposed training-free method can even outperform baselines with fine-tuning.

Despite different architectures of VLMs (Team, 2024; Bai et al., 2025; Guo et al., 2025), the leading performance is achieved by architectures employing a separate vision encoder to obtain vision tokens (Bai et al., 2025). In that architecture, both vision tokens and text tokens are available for token selection before applying LLMs. However, most of the existing work relies on *unimodality* for pruning while the multimodal information has not been explored sufficiently (Zhang et al., 2024; Yang et al., 2025a; Alvar et al., 2025). For example, SparseVLM (Zhang et al., 2024) mainly considers text tokens from language instruction to guide the pruning of vision tokens, while VisionZip (Yang et al., 2025a) heavily depends on the `[CLS]` vision token to select informative vision tokens. By investigating vision-language tasks, we find that given the same image, the answers can be different due to user-specific text queries, while the same text instruction can be applied for different images, i.e., caption tasks. Therefore, a unimodal method is hard to capture sufficient information about target tasks, implying a suboptimal performance for token selection.

In order to leverage both vision and text information to obtain informative vision tokens, in this work, we propose a multimodal strategy for efficient inference. First, we formulate the token selection problem as a **maximum coverage problem**, which aims to cover the target tokens with a subset of source tokens. While the source tokens are vision-only, the target ones can come from either text or vision, respectively. Therefore, the framework can explicitly combine the information from different modalities. Then, we optimize the coverage problem by maximizing a submodular function defined on the similarity between target and source tokens. Although the original problem is NP-hard (Khuller et al., 1999), a simple greedy algorithm can observe an approximate solution that is not worse than $(1 - 1/e)$ of the optimal solution (Nemhauser et al., 1978). The main contributions of this work are summarized as follows.

- We introduce the maximum coverage problem for vision token selection. The problem can be formulated as maximizing a submodular function, which has an efficient algorithm to obtain a near-optimal solution with a theoretical guarantee.
- We apply the coverage criterion to cover both the text tokens and the entire set of vision tokens with a subset of selected vision tokens. The text-vision and vision-vision coverage explicitly help explore multimodal information for selection.
- Experiments are conducted on benchmark datasets with diverse VLMs. The superior performance of the proposed method demonstrates the effectiveness of the proposed coverage criterion for the subset selection of vision tokens. For example, the proposed MMTok can achieve overall best performance under different settings as illustrated in Figure 1 (b) and shows the potential to compress to an extremely small number of vision tokens as in Figure 1 (a).

# 2 RELATED WORK

VLMs, such as LLaVA (Liu et al., 2023), InstructBLIP (Dai et al., 2023), and Qwen (Bai et al., 2025), have become a cornerstone for multimodal understanding by integrating large-scale vision encoders

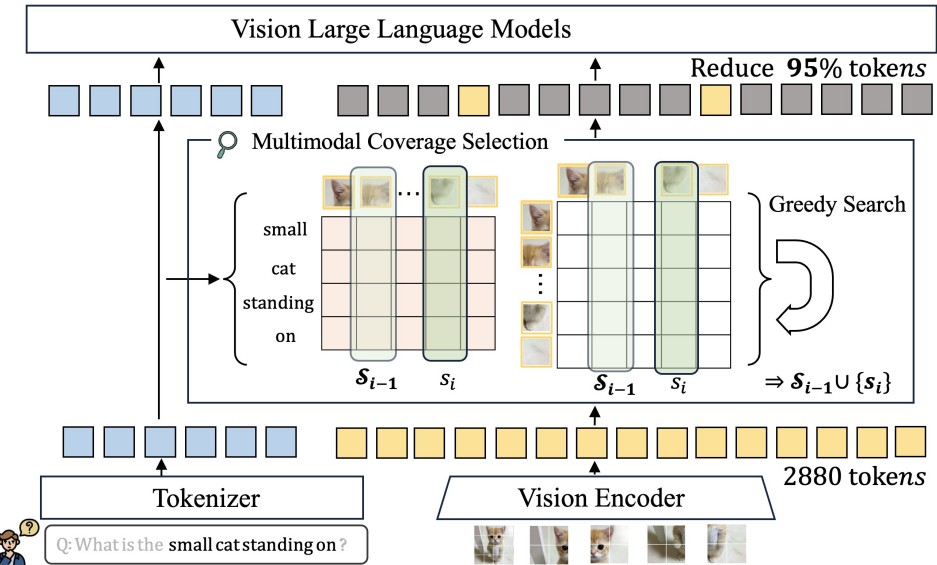

Figure 2: **Overview of MMTok framework.** Our method optimizes two maximum coverage problems simultaneously to leverage text-vision and vision-vision similarity for vision token selections.

(e.g., CLIP-ViT (Radford et al., 2021b)) with pre-trained language models. These models achieve strong performance by representing images as sequences of visual tokens, but their inference cost grows quadratically with token count, highlighting the need for more efficient processing.

Many vision token selection methods have been proposed recently, but most of them rely only on unimodal information for pruning (Yang et al., 2025a; Shang et al., 2024; Chen et al., 2024a; Zhang et al., 2024; Alvar et al., 2025). For example, VisionZip (Yang et al., 2025a) and FastV (Chen et al., 2024a) prune tokens using pre-trained attention signals, either ranking by [CLS] token attention (VisionZip) or discarding low-attention vision tokens in deeper layers (FastV). Besides ranking, DivPrune (Alvar et al., 2025) uses a diversity-based criterion but only has vision tokens to maximize the intra-set diversity. These methods rely on vision information and may miss query-related semantics (Jain & Wallace, 2019; Wiegreffe & Pinter, 2019). SparseVLM (Zhang et al., 2024) instead uses text-to-vision attention for scoring, yet ignores the information from the whole image. To mitigate the gap between existing unimodal algorithms and target multimodal tasks, we propose a coverage-based criterion to leverage both vision and text information sufficiently to select vision tokens effectively.

# 3 THE PROPOSED METHOD

To leverage the power of pre-trained models, many existing VLMs adopt a pre-trained vision encoder to extract vision tokens from images and then concatenate them with text tokens as input for the pre-trained LLMs. Although the simple architecture demonstrates promising performance, the inference efficiency can be challenging. Concretely, given an image, a pre-defined number of vision tokens will be obtained as $\{\mathbf{v}_1, \ldots, \mathbf{v}_n\}$. Even for a small $336 \times 336$ image, $n$ is 576 with the ViT-L-336px from CLIP (Radford et al., 2021a), which is much larger than that of the text tokens from the text query (Liu et al., 2023). The large $n$ will significantly slow down the inference of LLMs, which relies on the self-attention operations, and the complexity is quadratic to the total number of tokens.

To accelerate the inference of VLMs, we propose to select an informative subset of vision tokens $\{v_s\}_{s \in \mathcal{S}}$ to reduce the number of input tokens for LLM in VLM, where $\mathcal{N} = \{1, \ldots, n\}, \mathcal{S} \subseteq \mathcal{N}$, and $|\mathcal{S}| \ll n$. Figure 2 illustrates the framework of our method, and we describe it as follows.

## 3.1 VISION TOKEN SELECTION BY COVERAGE MAXIMIZATION

Unlike most of the existing work, we apply coverage as the main criterion for token selection. Given a similarity matrix $M \in \mathbb{R}^{m,n}$ defined between target tokens and source tokens, where $m$ denotes

the number of target tokens and $n$ is the number of source tokens, a subset $\mathcal{S}$ will be selected to maximize the similarity between the target and selected tokens as

$$f(\mathcal{S}; M) = \frac{1}{m} \sum_{i=1}^{m} \max M_{i,\mathcal{S}}; \quad \mathcal{S}^* = \arg\max_{\mathcal{S}} f(\mathcal{S}; M) \tag{1}$$

a.k.a. covering the target tokens by an appropriate subset of source tokens. We first find that Eq. 1 is a popular submodular function (Leskovec et al., 2007).

**Proposition 1.** *(Leskovec et al., 2007) For all subsets $\mathcal{A} \subseteq \mathcal{B} \subseteq \mathcal{N}$ and $s \in \mathcal{N} \setminus \mathcal{B}$,*

$$f(\mathcal{A} \cup \{s\}) - f(\mathcal{A}) \geq f(\mathcal{B} \cup \{s\}) - f(\mathcal{B})$$

Maximizing submodular functions in general is NP-hard (Khuller et al., 1999), but a simple greedy algorithm can achieve a good approximation.

**Proposition 2.** *(Nemhauser et al., 1978) Let $\mathcal{S}$ denote the subset obtained by the greedy algorithm, then we have*

$$f(\mathcal{S}) \geq (1 - 1/e) \max_{\mathcal{A}:|\mathcal{A}|=|\mathcal{S}|} f(\mathcal{A})$$

We elaborate on how to apply the coverage function for token selections in the following subsections.

### 3.1.1 MAXIMUM TEXT-VISION COVERAGE

First, we consider covering the semantics from text tokens with source vision tokens, which aims to find the vision tokens related to the text input (e.g., query). Let $\{\mathbf{t}_1, \ldots, \mathbf{t}_m\}$ denote the text tokens from the query. A similarity matrix between text and vision tokens can be obtained as

$$M_{i,j}^{tv} = \mathbf{t}_i^\top \mathbf{v}_j$$

where $M^{tv} \in \mathbb{R}^{m \times n}$ and $\forall i, j, \|\mathbf{t}_i\|_2 = \|\mathbf{v}_j\|_2 = 1$. To align the semantic similarity between text and vision, we adopt the vision tokens after the projection layer (i.e., those concatenated with text tokens as input for LLMs). After obtaining the appropriate similarity matrix, a subset of vision tokens can be selected to maximize the similarity between all text tokens and selected vision tokens for coverage as

$$\mathcal{S}' = \arg\max_{\mathcal{S}} f(\mathcal{S}; M^{tv})$$

According to Proposition 2, a greedy algorithm as summarized in Alg. 1 can approximate the optimal solution. It should be noted that the proposed Alg. 1 contains only simple operations (e.g., argmax, matrix multiplication, etc.) and thus is efficient for implementation.

---

**Algorithm 1** A Greedy Algorithm to Cover Text Input with Vision Tokens

1: **Input:** Similarity Matrix $M^{tv}$, $k$
2: Initialize $\mathcal{S} = \emptyset$
3: **for** $i = 1, \cdots, k$ **do**
4:     **for** $s \in \mathcal{N} \setminus \mathcal{S}$ **do**
5:         Compute $g(s) = f(\mathcal{S} \cup s; M^{tv})$
6:     **end for**
7:     Obtain $s_i = \arg\max_s g(s)$
8:     $\mathcal{S} = \mathcal{S} \cup s_i$
9: **end for**
10: **return** $\mathcal{S}$

---

**Algorithm 2** MMToK: A Greedy Algorithm for Multimodal Coverage

1: **Input:** Similarity Matrices $M^{tv'}$, $M^{vv'}$, $k$
2: Initialize $\mathcal{S} = \emptyset$
3: **for** $i = 1, \cdots, k$ **do**
4:     **for** $s \in \mathcal{N} \setminus \mathcal{S}$ **do**
5:         Compute $g(s) = f(\mathcal{S} \cup s; M^{tv'}, M^{vv'})$
6:     **end for**
7:     Obtain $s_i = \arg\max_s g(s)$
8:     $\mathcal{S} = \mathcal{S} \cup s_i$
9: **end for**
10: **return** $\mathcal{S}$

---

### 3.1.2 MAXIMUM VISION-VISION COVERAGE

Although text-vision coverage can explore vision information according to text, it may be insufficient due to vague text, e.g., "Please describe the image". Therefore, we propose to cover all vision information with a limited number of vision tokens. Concretely, a vision-vision similarity matrix can be generated as

$$M_{i,j}^{vv} = \mathbf{v}_i'^\top \mathbf{v}_j'$$

where $M^{vv} \in \mathbb{R}^{n \times n}$. Unlike $M^{tv}$ that adopts vision tokens after the projection layer to align with text tokens, those before projection are more appropriate to capture similarity between vision tokens without mixing text information. We have $\mathbf{v}'$ to distinguish it from the one after projection (i.e., $\mathbf{v}$).

Then, we can apply the greedy algorithm to select a subset of vision tokens to cover the main information implied by the whole set of vision tokens. Obviously, vision-vision coverage is complementary to text-vision coverage, which is also confirmed by our ablation study. The remaining challenge is to combine the two maximum coverage problems, which is described in the next subsection.

### 3.1.3 MAXIMUM MULTIMODAL COVERAGE

The maximum coverage problem can be applied to the original text and vision tokens simultaneously. However, $M^{tv}$ and $M^{vv}$ have different shapes and similarity measurements. Therefore, their values must be aligned before fusion. To calibrate the similarity between different modalities, the score for each row, i.e., that for target tokens, is first normalized by a softmax operation as

$$M_{i,j}^{tv'} = \frac{\exp(M_{i,j}^{tv}/\tau_t)}{\sum_{j=1}^{n} \exp(M_{i,j}^{tv}/\tau_t)}; \quad M_{i,j}^{vv'} = \frac{\exp(M_{i,j}^{vv}/\tau_v)}{\sum_{j=1}^{n} \exp(M_{i,j}^{vv}/\tau_v)}$$

where the softmax operation normalizes each row to a distribution over all vision tokens. $\tau_t$ and $\tau_v$ aim to further normalize the distribution shape for text-vision and vision-vision, respectively.

After calibration, the final objective for multimodal coverage can be written as

$$f(\mathcal{S}; M^{tv'}, M^{vv'}) = f(\mathcal{S}; M^{tv'}) + \alpha f(\mathcal{S}; M^{vv'}) \tag{2}$$

where $\alpha$ is used to weigh the importance of vision-vision coverage. Incorporating text-vision coverage with vision-vision coverage, the function in Eqn. 2 is still a submodular function as follows.

**Corollary 1.** *The sum of two submodular functions is a submodular function.*

*Proof.* It comes from the addition property of inequalities directly. ∎

With Corollary 1, we can apply a similar greedy algorithm to obtain the near-optimal solution for the multimodal scenario efficiently. The detailed algorithm is summarized in Alg. 2.

## 4 EXPERIMENTS

To evaluate the performance of the proposed method, MMTok, we conduct experiments on diverse benchmark datasets and VLMs with different architectures. For a fair comparison, we use datasets adopted in VisionZip (Yang et al., 2025a), which contains GQA (Hudson & Manning, 2019), MM-Bench (Liu et al., 2024c), MME (Fu et al., 2023), POPE (Li et al., 2023b), ScienceQA(IMG) (Lu et al., 2022), VQAv2-Test-Dev (Goyal et al., 2017), TextVQA (Singh et al., 2019), MMMU (Yue et al., 2024), and SeedBench (Li et al., 2023a). Meanwhile, five VLMs are applied for comparison, that is, LLaVA-1.5-7B (Liu et al., 2023), LLaVA-1.5-13B (Liu et al., 2023), LLaVA-NeXT-7B (Liu et al., 2024a), LLaVA-NeXT-13B (Liu et al., 2024a), and a recent model Qwen-2.5-VL-7B (Bai et al., 2025). Finally, we compare our method with state-of-the-art vision token pruning algorithms, including FastV (Chen et al., 2024a) (a vision-based method), SparseVLM (Zhang et al., 2024) (a language-based method), VisionZip (Yang et al., 2025a) (a [CLS]-importance-based method), and DivPrune (Alvar et al., 2025) (a diversity-based method). We also include a fine-tuning-based method, VisionZip🔥, in the comparison. We obtain the result of DivPrune through its official code, and that for other baselines is directly from (Yang et al., 2025a). Evaluation is implemented under the `Lmms-eval` framework (Li et al., 2024a) with the details elaborated as follows.

**Implementation Details** The proposed method relies on an appropriate similarity for coverage optimization. Since different layers may demonstrate different similarity measurements (Liu et al., 2023), we have vision tokens before the projection layer to compute vision-vision similarity, while those after the projection layer are for text-vision coverage. That is because the latter layer aligns better with the text. We find that our method is not sensitive to hyperparameters, as shown in the ablation study. Therefore, we fix $\tau_t = 0.02$, $\tau_v = 0.2$, and $\alpha = 0.5$ for all experiments if not otherwise specified.

| Method | GQA | MMB | MME | POPE | SQA | VQA$^{V2}$ | VQA$^{Text}$ | MMMU | SEED | Avg. |
|---|---|---|---|---|---|---|---|---|---|---|
| | | | | *Total 576 Tokens* | | | | | | |
| LLaVA-1.5-7B | 61.90 | 64.70 | 1862.00 | 85.90 | 69.50 | 78.50 | 58.20 | 36.30 | 58.60 | 100% |
| | | | | *Retain 192 Tokens* ↓ 67% | | | | | | |
| FastV | 52.70 | 61.20 | 1612.00 | 64.80 | 67.30 | 67.10 | 52.50 | 34.30 | 57.10 | 89.6% |
| SparseVLM | 57.60 | 62.50 | 1721.00 | 83.60 | 69.10 | 75.60 | 56.10 | 33.80 | 55.80 | 95.5% |
| VisionZip | 59.30 | 63.00 | 1782.60 | 85.30 | 68.90 | 76.80 | 57.30 | 36.60 | 56.40 | 97.9% |
| DivPrune | 59.97 | 62.54 | 1762.23 | 87.00 | 68.66 | 76.87 | 56.97 | 35.44 | 58.71 | 98.0% |
| VisionZip🔥 | 60.10 | 63.40 | 1834.00 | 84.90 | 68.20 | 77.40 | 57.80 | 36.20 | 57.10 | 98.4% |
| MMTok | 60.07 | 63.40 | 1773.86 | 86.42 | 68.76 | 77.11 | 57.68 | 36.33 | 59.21 | **98.7%** |
| | | | | *Retain 128 Tokens* ↓ 78% | | | | | | |
| FastV | 49.60 | 56.10 | 1490.00 | 59.60 | 60.20 | 61.80 | 50.60 | 34.90 | 55.90 | 84.4% |
| SparseVLM | 56.00 | 60.00 | 1696.00 | 80.50 | 67.10 | 73.80 | 54.90 | 33.80 | 53.40 | 92.9% |
| VisionZip | 57.60 | 62.00 | 1761.70 | 83.20 | 68.90 | 75.60 | 56.80 | 37.90 | 54.90 | 96.8% |
| DivPrune | 59.25 | 62.03 | 1718.22 | 86.72 | 68.66 | 75.96 | 56.06 | 35.56 | 56.98 | 96.9% |
| VisionZip🔥 | 58.90 | 62.60 | 1823.00 | 83.70 | 68.30 | 76.60 | 57.00 | 37.30 | 55.80 | 97.7% |
| MMTok | 59.29 | 62.29 | 1779.14 | 86.25 | 68.82 | 76.35 | 57.03 | 35.67 | 58.59 | **97.8%** |
| | | | | *Retain 64 Tokens* ↓ 89% | | | | | | |
| FastV | 46.10 | 48.00 | 1256.00 | 48.00 | 51.10 | 55.00 | 47.80 | 34.00 | 51.90 | 75.6% |
| SparseVLM | 52.70 | 56.20 | 1505.00 | 75.10 | 62.20 | 68.20 | 51.80 | 32.70 | 51.10 | 86.9% |
| VisionZip | 55.10 | 60.10 | 1690.00 | 77.00 | 69.00 | 72.40 | 55.50 | 36.20 | 52.20 | 93.2% |
| DivPrune | 57.78 | 59.28 | 1674.40 | 85.56 | 68.07 | 74.11 | 54.69 | 35.56 | 55.13 | 94.8% |
| VisionZip🔥 | 57.00 | 61.50 | 1756.00 | 80.90 | 68.80 | 74.20 | 56.00 | 35.60 | 53.40 | 95.0% |
| MMTok | 58.29 | 61.17 | 1715.33 | 85.77 | 69.16 | 75.20 | 56.01 | 36.11 | 57.15 | **96.6%** |

Table 1: **Performance Comparison on LLaVA-1.5-7B.** More details in Appendix Table 16.

| Method | LLaVA-1.5-7B (2023) | | | LLaVA-1.5-13B (2023) | | | LLaVA-NeXT-7B (2024a) | | | LLaVA-NeXT-13B (2024a) | | |
|---|---|---|---|---|---|---|---|---|---|---|---|---|
| | 576 tokens | | | 576 tokens | | | Upper(Up.) 2880 tokens | | | Upper(Up.) 2880 tokens | | |
| Compress Ratio | ↓ 67% | ↓ 78% | ↓ 89% | ↓ 67% | ↓ 78% | ↓ 89% | ↓ 78% | ↓ 89% | ↓ 94% | ↓ 78% | ↓ 89% | ↓ 94% |
| Remain Token | 192 | 128 | 64 | 192 | 128 | 64 | Up. 640 | Up. 320 | Up. 160 | Up. 640 | Up. 320 | Up. 160 |
| VisionZip | 97.9% | 96.8% | 93.2% | 97.9% | 97.0% | 93.7% | 97.5% | 94.5% | 90.4% | 97.7% | 94.7% | 91.4% |
| DivPrune | 98.0% | 96.9% | 94.8% | 98.2% | 96.9% | 95.3% | 97.1% | 95.1% | 92.4% | 97.1% | 94.5% | 92.0% |
| VisionZip🔥 | 98.4% | 97.7% | 95.0% | 98.7% | 97.4% | 94.8% | 98.9% | 97.6% | 95.0% | 98.8% | 97.8% | 94.6% |
| MMTok | **98.7%** | **97.8%** | **96.6%** | **98.7%** | **97.5%** | **96.4%** | **98.7%** | **97.3%** | **95.1%** | **98.2%** | **96.4%** | **95.1%** |

Table 2: **Comparison on LLaVA-1.5 and LLaVA-NeXT.** Details are in Appendix Tables 16 to 19.

## 4.1 PERFORMANCE COMPARISON ON DIVERSE TASKS

**LLaVA-1.5-7B** First, we compare our method with baselines using LLaVA-1.5-7B, which is a popular benchmark for vision token selection. The model has a fixed number of vision tokens for arbitrary visual inputs. As shown in Table 1, given the original 576 tokens, MMTok achieves the best performance (preserving on average 98.7/97.8/96.6% original performance of LLaVA-1.5-7B), when retaining only 192/128/64 tokens (reducing by 67/78/89% of tokens compared to 576), respectively. Specifically, our method outperforms DivPrune by 1.8% when using a budget of 64 tokens. Although the gap decreases with more tokens as expected, MMTok still surpasses all baselines without fine-tuning by at least 0.7% with 192 tokens. In addition, compared to the fine-tuning method, the proposed method is 1.6% better than VisionZip🔥 with 64 tokens, which shows the potential of the training-free strategy. Since VisionZip and DivPrune show much better performance than FastV and SparseVLM, we will include only them for comparison in the following experiments.

**LLaVA-1.5-13B** The average performance over all benchmark datasets and token budgets is reported in Table 2, while detailed results can be found in Appendix Table 17. Although the model is larger, the observation is similar to the above 7B counterpart, where our method consistently outperforms the baselines with a clear margin.

**LLaVA-NeXT 7B and 13B** In addition to models that have a fixed number of vision tokens, we further evaluate our method on LLaVA-NeXT (Liu et al., 2024a), which dynamically samples up to five images and processes them individually, resulting in up to 2880 vision tokens. To align the comparison with real applications, we keep the dynamic settings as VisionZip (Yang et al., 2025a) and

have token selection performed in a fixed ratio. For example, with a maximum budget of 160 tokens, we retain 32 tokens per image in up to five images ($32 \times 5 = 160$). The retained number of tokens becomes 128 if only four images are sampled by the VLMs according to the ratio of $160/2,880$. The same setting is used for all baselines as a fair comparison. As shown in Table 2, our method retains more than 95% of the original performance using only 5.5% of the tokens with a budget of 160 tokens, indicating substantial redundancy in vision tokens and the effectiveness of the proposed strategy. Detailed results can be found in Appendix Tables 18 and 19.

| Method | GQA Acc. ↑ | MMB Acc. ↑ | MME P+C ↑ | POPE F1 ↑ | VQA$^{\text{Text}}$ Acc. ↑ | SQA Acc. ↑ | OCRBench Acc. ↑ | Avg.† ↑ |
|---|---|---|---|---|---|---|---|---|
| *Dynamic Resolution (MinPix = 256 × 28 × 28, MaxPix = 2048 × 28 × 28), Upper Bound* **(100%)** | | | | | | | | |
| Avg. Tokens $\bar{T}$ | 358.5 | 276.9 | 867.6 | 359.6 | 976.5 | 323.0 | 652.8 | |
| Qwen-2.5-VL-7B | 60.48 | 83.25 | 2327 | 86.16 | 77.72 | 87.46 | 83.80 | 100% |
| *Fixed Resolution (MinPix = MaxPix = 2048 × 28 × 28), Upper Bound* **(100%)** | | | | | | | | |
| Qwen-2.5-VL-7B | 58.59 | 83.59 | 2339 | 86.09 | 76.64 | 86.91 | 76.60 | 99.3% |
| *Retain 20%* $\bar{T}$ | 71.7 | 55.4 | 173.5 | 71.9 | 195.3 | 64.6 | 130.6 | ↓ **80%** |
| VisionZip | 56.80 | 80.33 | 2174 | 83.38 | 70.43 | 84.23 | 59.50 | 94.2% |
| DivPrune | 56.70 | 76.98 | 2163 | 80.59 | 65.86 | 80.91 | 48.10 | 91.5% |
| MMTok | 58.09 | 79.30 | 2217 | 82.38 | 70.49 | 81.61 | 59.60 | **94.6%** |
| *Retain 10%* $\bar{T}$ | 35.9 | 27.7 | 86.8 | 36.0 | 97.7 | 32.3 | 65.3 | ↓ **90%** |
| VisionZip | 52.47 | 75.60 | 2003 | 78.90 | 63.78 | 82.30 | 36.90 | 87.5% |
| DivPrune | 53.43 | 72.85 | 1957 | 74.99 | 59.59 | 79.57 | 37.30 | 84.7% |
| MMTok | 55.09 | 74.74 | 2051 | 78.75 | 63.90 | 80.47 | 43.60 | **88.5%** |
| *Retain 5%* $\bar{T}$ | 17.9 | 13.8 | 43.4 | 18.0 | 48.8 | 16.2 | 32.6 | ↓ **95%** |
| VisionZip | 46.28 | 67.53 | 1677 | 66.38 | 54.49 | 79.57 | 19.70 | 75.4% |
| DivPrune | 49.01 | 65.89 | 1739 | 68.45 | 52.02 | 77.05 | 24.90 | 76.3% |
| MMTok | 50.66 | 65.89 | 1796 | 71.35 | 55.95 | 77.19 | 30.70 | **79.0%** |
| *0 Token* ↓ **100%** | | | | | | | | |
| Qwen-2.5-VL-7B | 31.84 | 20.10 | 935 | 0.00* | 38.93 | 71.10 | 1.80 | 33.8% |

Table 3: **Comparison on Qwen-2.5-VL-7B.** Avg.† are computed over 5 datasets. *When no visual tokens are provided, Qwen-2.5-VL outputs `"No"` for all questions, leading to 0% F1. More detailed results are in Appendix Table 20.

**Qwen-2.5-VL-7B** Finally, we compare different algorithms on a more advanced VLM, that is, Qwen-2.5-VL-7B. Unlike previous work, it adopts dynamic resolution and a token-merging layer. Those strategies help reduce the total number of vision tokens while demonstrating better performance. For example, the average number of input tokens is only 359.6 in Qwen on POPE, significantly less than 2880 tokens in LLaVA-NeXT. Therefore, it is more challenging to apply the token selection algorithm in this strong model. Following experiments for LLaVA-NeXT, we conduct the evaluation under dynamic resolution for all methods. Due to distinct image pre-processing strategies in Qwen, we include 7 image datasets in this comparison. Since ScienceQA (SQA) is a low-IC dataset that will be discussed in Section 4.2 and all baselines perform poorly on OCRBench, the average performance is computed across the remaining 5 datasets. For MMTok, we reduce $\tau_t$ to 0.01 for all datasets while other parameters remained.

First, we compare the dynamic resolution to the fixed number of tokens in Qwen as shown in the first two rows of Table 3. Although the model can use a fixed number of about 2,048 vision tokens for different tasks, the performance is worse than that of the dynamic strategy, which has much fewer tokens. It shows that vision tokens are quite redundant for VLM tasks, and the sophisticated strategies in Qwen already compress the number to hundreds, providing even better performance. Based on the challenging dynamic setting, we further investigate whether token selection is still valuable. From Table 3, we can find that our MMTok can preserve nearly 95% of the original performance while further reducing the number of vision tokens to 20%. This observation demonstrates that even for models with token compression, the remaining tokens can still be redundant. The proposed method MMTok can effectively explore the most informative tokens and further reduce the number of vision tokens from hundreds to tens. Furthermore, our method is better than VisionZip with different budgets, which confirms the efficacy of our proposed multimodal coverage strategy. Finally, we can observe that even without any vision tokens, Qwen's performance on SQA is still close to its version with all tokens. This reminds us to investigate the contribution of vision to vision-language tasks in the next subsection, which can help to better evaluate the performance of token selection methods.

## 4.2 Comparison on High IC Tasks with Limited Vision Tokens

Although multimodal tasks rely on images for answers, the contribution of vision varies. Table 4 summarizes the performance with/without vision tokens on different datasets. It is interesting to observe that even without any vision tokens, LLaVA-1.5 still preserves 92% of the original performance on MMMU and 91% on ScienceQA. Those tasks may fail to help adequately assess the efficacy of vision token selection. To mitigate the issue, we introduce **Image Contribution (IC)** to quantify the relative performance gain from all vision tokens, $IC = (\text{Perf}_{\text{All}} - \text{Perf}_0)/\text{Perf}_0$ and summarize IC values in Table 4. According to the table, we can identify 5 and 6 high-IC datasets for LLaVA and LLaVA-NeXT, respectively. Then, we compare different algorithms on those datasets in Table 5. To evaluate the performance with an extremely aggressive compression ratio, we extend the experiments from 64 tokens to 2 tokens. The comparison shows that our method can substantially preserve the informative vision tokens for VL tasks. Moreover, we illustrate the performance ratio compared to the original result in Figure 1. On POPE, our method maintains about 80% original performance with only 2 vision tokens, showing the importance of appropriate vision tokens. More results can be found in Appendix Tables 21 and 22.

| Dataset | LLaVA-1.5-7B | | LLaVA-NeXT-7B | |
|---|---|---|---|---|
| | All / Zero | IC | All / Zero | IC |
| MMB | 64.7/19.33 | **2.347** | 67.9/17.87 | **2.801** |
| POPE | 85.9/44.64 | 0.924 | 86.4/25.84 | 2.344 |
| MME | 1862/970.89 | 0.918 | 1842/867 | 1.125 |
| SEED-I | 66.14/37.03 | 0.786 | 70.2/37.43 | 0.875 |
| GQA | 61.9/37.65 | 0.644 | 64.2/38.23 | 0.679 |
| TextVQA | 58.2/41.66 | 0.397 | 61.3/37.77 | 0.623 |
| SQA | 69.5/63.51 | 0.094 | 70.2/64.60 | 0.087 |
| MMMU | 36.3/33.33 | 0.089 | 35.1/31.56 | 0.112 |

Table 4: Demonstration of Image Contribution (IC).

| Model/Method | Different Token Budgets | | | | | |
|---|---|---|---|---|---|---|
| | 64 | 32 | 16 | 8 | 4 | 2 |
| *LLaVA-1.5-7B (MMB, POPE, MME, SEED, GQA)* | | | | | | |
| VisionZip | 90.0% | 83.5% | 69.7% | 48.9% | 43.8% | 43.0% |
| DivPrune | 93.1% | 89.6% | 81.4% | 68.4% | 54.3% | 50.1% |
| MMTok (Ours) | **94.7%** | **91.0%** | **86.4%** | **79.8%** | **71.4%** | **62.1%** |
| *LLaVA-NeXT-7B (MMB, POPE, MME, SEED, GQA, TextVQA)* | | | | | | |
| VisionZip | 93.2% | 87.5% | 76.8% | 52.6% | 39.4% | 38.5% |
| DivPrune | 94.2% | 89.7% | 85.7% | 79.9% | 71.0% | 57.8% |
| MMTok (Ours) | **95.7%** | **92.8%** | **89.6%** | **85.2%** | **79.2%** | **71.0%** |

Table 5: Comparison on high-IC tasks with different token budgets.

## 4.3 Ablation Study

We conduct comprehensive ablation studies to demonstrate each component in MMTok. All experiments are performed on LLaVA-1.5-7B with 64 tokens unless otherwise specified.

| Multimodal Coverage | GQA Acc. ↑ | MMB Acc. ↑ | MME P+C ↑ | POPE F1 ↑ | SQA Acc. ↑ | VQA$^{\text{Text}}$ Acc. ↑ | MMMU Acc. ↑ | SEED Acc. ↑ | Avg. ↑ |
|---|---|---|---|---|---|---|---|---|---|
| *Total 576 Tokens* **(100%)** | | | | | | | | | |
| LLaVA-1.5-7B | 61.9 | 64.7 | 1862 | 85.9 | 69.5 | 58.2 | 36.3 | 58.6 | 100.0% |
| *Retain 64 Tokens* ↓ 88.9% | | | | | | | | | |
| T-V ($M^{tv}$) | 56.82 | 59.62 | 1632.47 | 83.56 | _68.72_ | 51.97 | 35.33 | 56.36 | 93.8% |
| V-V ($M^{vv}$) | _58.14_ | 59.88 | 1662.34 | 83.43 | 67.67 | 53.93 | 35.33 | _56.90_ | 94.7% |
| Softmax T-V ($M^{tv'}$) | 56.66 | 58.85 | 1674.11 | 83.69 | 68.57 | 52.01 | 35.33 | 56.37 | 93.9% |
| Softmax V-V ($M^{vv'}$) | 57.97 | _60.31_ | _1684.33_ | _84.31_ | 68.07 | _55.90_ | _35.89_ | 56.88 | _95.7%_ |
| MMTok ($M^{tv'} + M^{vv'}$) | **58.29** | **61.17** | **1715.33** | **85.77** | **69.16** | **56.01** | **36.11** | **57.15** | **96.7%** |

Table 6: **Ablation on multimodal coverage in MMTok**. The best performance with token selection is highlighted in bold and the second-best is underlined.

**Unimodal Coverage vs. Multimodal Coverage** The proposed method contains both text-vision coverage (T-V) and vision-vision coverage (V-V). We evaluate each component separately in Table 6. Compared with the original similarity matrix, the softmax variant can obtain a similar or even better performance, which shows that calibration on similarity matrices will not hurt the performance. Then, combining coverage optimization on different modalities shows an improvement of about 1% over unimodal coverage, which demonstrates the complementarity between T-V and V-V coverages for various tasks. More ablation experiments can be found in the Appendix.

**Token Selection vs. Resizing** Resizing the image resolution is an effective way to reduce the total number of tokens as shown in (Yang et al., 2025b). We compare the resize strategy with token

budgets in Table 7. First, we can find that selecting vision tokens from the original large image can be more effective than resizing under the same token budget. This is because resizing would ignore the redundancy between vision tokens. More importantly, we observe that incorporating with resizing, MMTok can work better than the counterpart with the same token budget on full images. For example, with about 10% original tokens, MMTok with resizing can achieve 2170 on MME while that on original image is only 2051. Exploring resizing with token selection sufficiently can be our future work.

| Model | Image Resize Ratio | Token Avg. | MME P+C ↑ |
|---|---|---|---|
| Qwen2.5-VL-7B | 1 | 867.6 | 2327 |
| MMTok | 1 | 173.5 | 2217 |
| MMTok | 1 | 86.8 | 2051 |
| Resize image to fixed ratio of original height and width, respectively | | | |
| Qwen2.5-VL-7B | 1/2 | 459.1 | 2274 |
| Qwen2.5-VL-7B | 1/4 | 349.6 | 2238 |
| Qwen2.5-VL-7B | 1/8 | 276.0 | 1793 |
| Retain 55% Tokens on 1/4 Resized Image | | | |
| MMTok | 1/4 | 192.3 | 2254 |
| Retain 40% Tokens on 1/4 Resized Image | | | |
| MMTok | 1/4 | 139.8 | 2215 |
| Retain 25% Tokens on 1/4 Resized Image | | | |
| MMTok | 1/4 | 87.4 | 2170 |

Table 7: Comparison with resize strategy on MME with Qwen2.5-VL-7B. The original image is denoted as resize ratio of 1. Qwen has a default minimal number of vision tokens as 256 to obtain meaningful results.

| Model | Upper Token | Total Infer T(s) | POPE Infer T(s) | GPU Util. | Memory (+25.42 GB) | POPE F1 | SEED Acc. | TextVQA Acc. | MME P+C | MMB Acc. | GQA Acc. | Avg. (%) |
|---|---|---|---|---|---|---|---|---|---|---|---|---|
| | | | | | *H100 Single GPU Performance, Upper 2880 Tokens* | | | | | | | |
| LLaVA-NeXT-13B | 2880 | 15204 | 1705 | 86.7% | 4.59 | 86.22 | 71.89 | 64.33 | 1900.86 | 69.16 | 65.38 | 100.0 |
| VisionZip | Upper 160 | 7551 | 866 | 52.4% | 1.92 | 76.32 | 61.18 | 58.33 | 1738.24 | 64.78 | 57.77 | 89.6 |
| DivPrune | Upper 160 | 8186 | 1060 | 50.9% | 1.23 | 82.16 | 63.80 | 54.65 | 1699.83 | 64.78 | 59.34 | 90.5 |
| MMTok | Upper 160 | 7768 | 913 | 58.0% | 1.78 | 85.11 | 65.45 | 55.91 | 1811.35 | 65.89 | 61.94 | 93.7 |

Table 8: **Comparison of Inference Efficiency.** All results are reproduced under the same hardware and evaluation settings. The initial memory usage for loading the model is 25.42GB.

**Inference Efficiency** Besides effectiveness, we examine efficiency in real scenarios. To mimic real applications, we report the total running time on different datasets in Table 8. First, we profile the computational cost on POPE. Obviously, all token selection methods help reduce the utility percentage of the GPU by about 30%, which shows that pruning is helpful for inference. Then, with a fixed memory cost of 25.42GB for model loading, these methods can also help reduce the usage of running-time memory by more than 58.2% compared to the baseline. This reduction in computation and memory helps significantly improve the inference time on POPE, where both VisionZip and our method can reduce the running time by about 50%. DivPrune runs a little bit slower due to a different strategy for handling multiple crops in its official code. Although our method introduces two subproblems, that is, T-V and V-V to optimize, the running time is almost the same as the fast unimodal method, i.e., VisionZip, which confirms the efficiency of MMTok. The running time accumulated over 6 tasks demonstrates a similar observation, where the performance of MMTok is better than VisionZip by 4.1%. This further demonstrates the efficacy and efficiency of our proposal.

**Visualization** While MMTok will leverage text information for vision token selection, the vision-vision coverage in our framework can help preserve the vision information and reuse the selected vision tokens for multi-turn conversation as shown in Figure 3. We can observe that MMTok selects tokens with text from Q1 but vision-vision coverage enables the following questions to be answered correctly with the selected vision tokens.

In addition, we also demonstrate how the answer changes with the number of selected tokens in Figure 4. Given a simple question as in the first example, only 8 vision tokens can obtain the right answer. However, for the challenging question as in the last example, the answer changes even with 16 tokens. The phenomenon implies that the appropriate number of vision tokens also depends on the hardness of the questions. Hardness-aware vision token selection is an interesting future direction.

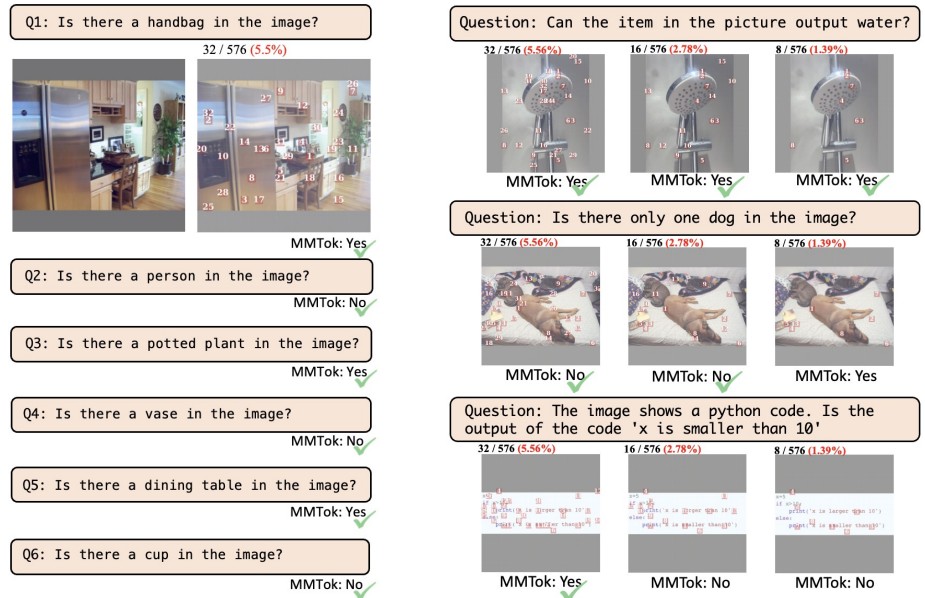

Figure 3: Multi-turn conversation by applying MMTok only with text from Q1.

Figure 4: Answer changes with different number of tokens. Hard questions need more vision tokens.

## 5 CONCLUSION

In this work, we propose a multimodal coverage framework, MMTok, to guide vision token selection to accelerate the inference of VLMs in a training-free manner. Extensive experiments on benchmark datasets and representative VLMs demonstrate that our method outperforms the unimodal baselines without compromising efficiency. While text input may carry limited semantic information as a target for vision tokens to cover, a lightweight agent VLM can be leveraged to provide additional meaningful text tokens to guide the selection of the vision tokens, which can be the future direction.

## 6 ETHICS STATEMENT

To the best of our knowledge, this work has no potential ethical issues to disclose.

## 7 REPRODUCIBILITY STATEMENT

To ensure the reproducibility of this work, all implementation details have been clearly described, and we have also released the code.

## 8 ACKNOWLEDGMENTS

The authors thank Dr. Yebowen Hu and Dr. Kaiqiang Song for their valuable discussions, as well as their respective assistance with experiments and computing resource scheduling. Dr. Juhua Hu is supported by Advata and EWA Gift Funding. Dr. Yanjie Fu is supported by the National Science Foundation (NSF) through grant numbers: 2426340, 2416727, 2421865, 2421803. All opinions, findings, conclusions, and recommendations in this work are those of the authors and do not necessarily reflect the views of the funding agencies.

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

## A   LLM USAGE STATEMENT

We did not use LLM at all during the idea and writing stage of this work.

## B   EXPERIMENTS

### B.1   ADAPTIVE TEMPERATURE $\tau_v^a$

To further improve the calibration between $M^{tv'}$ and $M^{vv'}$, an adaptive visual temperature can be applied for each example. Concretely, when fixing $\tau_t$, the maximal similarity between the target text tokens and the whole set of vision tokens can be obtained as $f(\mathcal{N}; M^{tv'})$, letting $\mathcal{S} = \mathcal{N}$. The desired temperature $\tau_v$ should lead to a similar magnitude for the vision-vision similarity. The optimization problem can be cast as

$$\min_{\tau_v^a} |f(\mathcal{N}; M^{tv'}) - f(\mathcal{N}; M^{vv'})|$$

For the default $f$, it is monotone to $\tau_v^a$, which can be solved efficiently by bisection search. However, the diagonal elements in $M^{vv'}$ can mislead the optimization due to their fixed value of 1. To mitigate the issue, the $k$-th largest value is applied to search for the temperature as

$$\min_{\tau_v^a} |f(\mathcal{N}; M^{tv'}) - f_k(\mathcal{N}; M^{vv'})|; \quad f_k(\mathcal{N}; M^{vv'}) = \frac{1}{n} \sum_{i=1}^{n} \max_k M_{i,:}^{vv'}$$

Moreover, $f_k$ is not guaranteed to be a monotone function to $\tau_v^a$, and we can search the value in $(\tau_t, \tau_v]$ as suggested in (Qian et al., 2023), where it shows that the temperature between vision-vision should be higher than that between text-vision due to the modality gap.

We perform the evaluation on high IC tasks in Table 9. As discussed above, the second largest value is adopted for searching the temperature in the set of $\{0.05, 0.1, 0.15, 0.2\}$. While the variant with adaptive temperature, i.e., MMTok$_{Adapt}$, shows a slightly better performance with a budget of 16 tokens, the results over different tasks are almost the same, demonstrating that our method is insensitive to hyperparameters.

| Method | GQA Acc. ↑ | MMB Acc. ↑ | POPE F1 ↑ | MME P+C ↑ | SEED Acc. ↑ | Avg. ↑ |
|---|---|---|---|---|---|---|
| *Upper Bound: LLaVA-1.5-7B (576 Tokens)* | | | | | | |
| LLaVA-1.5 7B | 61.9 | 64.7 | 85.9 | 1862 | 58.6 | 100% |
| *Retain 16 Tokens ↓ 97.2%* | | | | | | |
| MMTok | 53.31 | 54.30 | 79.79 | 1550.65 | 56.67 | 88.6% |
| MMTok$_{Adapt}$ | 53.31 | 54.30 | 79.83 | 1565.10 | 56.66 | 88.7% |

Table 9: **Fixed vs. Adaptive Temperature.** Evaluation on LLaVA-1.5 7B with adaptive temperature $\tau_v^a \in \{0.05, 0.1, 0.15, 0.2\}$.

### B.2   POOLING STRATEGY FOR TEXT

Given an LLM, each word can be tokenized into multiple tokens. To recover the semantic information of words, we may aggregate tokens from the same word. In this experiment, we explore different pooling strategies when computing T-V similarity. Concretely, we consider the pooling process either before or after computing the similarity matrix, where `Max`-pooling selects the token with the maximum feature value or similarity, `Mean`-pooling averages similarity over all tokens, and `First`-pooling simply retains the first token of a word. As shown in Table 10, there is no pooling strategy that consistently yields the best performance across all eight datasets. Therefore, our method does not apply word pooling for simplicity.

### B.3   MMTOK FOR REASONING TASK

To demonstrate the efficacy of MMTok on reasoning task, we conduct the experiments on the MMStar dataset (Chen et al., 2024b). In Table 11, we can observe that vision token selection can also help

| Pooling Method | Position | GQA Acc. ↑ | MMB Acc. ↑ | MME P+C ↑ | POPE F1 ↑ | SQA Acc. ↑ | VQA$^{Text}$ Acc. ↑ | MMMU Acc. ↑ | SEED Acc. ↑ | Avg. ↑ |
|---|---|---|---|---|---|---|---|---|---|---|
| *MMTok on LLaVA-1.5-7B with 64 Tokens (Baseline)* | | | | | | | | | | |
| None | - | 58.29 | 61.17 | **1715** | **85.77** | **69.16** | **56.01** | 36.11 | 57.15 | **100.0%** |
| *Pre-Pooling (Before Similarity Calculation)* | | | | | | | | | | |
| Mean | Pre | 58.01 | 61.00 | 1703 | 85.75 | 69.11 | 55.73 | 36.00 | 57.13 | 99.7% |
| Max | Pre | 58.26 | 61.17 | 1704 | 85.64 | 68.82 | 55.82 | 35.89 | 56.94 | 99.7% |
| First | Pre | **58.39** | **61.34** | 1709 | 85.67 | 68.77 | 55.76 | 36.11 | 56.90 | 99.8% |
| *Post-Pooling (After Similarity Calculation)* | | | | | | | | | | |
| Mean | Post | 58.20 | 61.00 | 1690 | 85.67 | 68.77 | 55.77 | **36.22** | **57.16** | 99.7% |
| Max | Post | 58.36 | 61.00 | 1711 | 85.61 | 68.77 | 55.68 | **36.22** | 57.04 | 99.8% |
| First | Post | **58.39** | **61.34** | 1709 | 85.67 | 68.77 | 55.76 | 36.11 | 56.90 | 99.8% |

Table 10: **Word token pooling strategies for token selection on LLaVA-1.5-7B.** Pre-pooling aggregates subword tokens before similarity computation, while post-pooling applies pooling afterward. We evaluate three methods: `Mean` (average pooling), `Max` (maximum pooling), and `First` (first subword). The baseline applies no pooling. The best is in bold and the second-best is underlined.

reasoning tasks. It should be noted that the major issue on these challenging tasks is that the original performance without selection is already quite low. Therefore, it is hard to show the significant difference when the upper-bound is limited. Nevertheless, our method is still better than baselines with a clear margin and by selecting 32 out of 576 tokens, MMTok is able to recover the performance of the baseline.

| Method | MMStar Metrics | | | | | | |
|---|---|---|---|---|---|---|---|
| | Coarse | Fine-Grained | Instance | Logical | Math | Sci&Tech | Average |
| *Baseline* | | | | | | | |
| LLaVA-1.5-7B | 63.63 | 25.63 | 38.89 | 28.92 | 26.60 | 18.48 | 33.69 |
| *64 Tokens* | | | | | | | |
| VisionZip | 55.27 | 22.92 | 39.32 | 27.95 | 24.76 | 24.62 | 32.47 |
| DivPrune | 56.35 | 19.50 | 36.72 | 27.73 | 26.70 | 18.94 | 30.99 |
| Ours | 59.08 | 22.66 | 39.51 | 29.66 | 28.39 | 20.66 | 33.33 |
| *32 Tokens* | | | | | | | |
| VisionZip | 48.58 | 19.04 | 39.73 | 29.69 | 22.91 | 21.95 | 30.32 |
| DivPrune | 54.82 | 21.07 | 37.03 | 27.82 | 24.18 | 19.32 | 30.71 |
| Ours | 59.56 | 25.71 | 40.49 | 29.94 | 27.92 | 17.37 | 33.50 |
| *16 Tokens* | | | | | | | |
| VisionZip | 43.76 | 21.34 | 32.58 | 25.97 | 23.18 | 19.96 | 27.80 |
| DivPrune | 49.99 | 21.45 | 38.37 | 28.45 | 21.54 | 18.58 | 29.73 |
| Ours | 56.32 | 21.58 | 39.48 | 30.22 | 23.98 | 15.16 | 31.12 |
| *8 Tokens* | | | | | | | |
| VisionZip | 27.93 | 21.26 | 25.71 | 21.84 | 20.18 | 17.83 | 22.46 |
| DivPrune | 47.25 | 21.05 | 33.89 | 25.75 | 20.32 | 16.76 | 27.50 |
| Ours | 54.11 | 21.26 | 35.09 | 29.56 | 20.34 | 15.04 | 29.23 |
| *0 Tokens* | | | | | | | |
| Baseline | 31.85 | 19.10 | 23.77 | 23.61 | 14.28 | 16.11 | 21.45 |

Table 11: Comparison on MMStar with LLaVA-1.5-7B.

## B.4 INFERENCE EFFICIENCY FOR QWEN2.5-VL-7B

Besides the evaluation in Table 8, we also evaluate the inference efficiency of Qwen2.5-VL-7B in Table 12 using the MME task. We find that vision token selection can also accelerate the inference of state-of-the-art VLMs.

| Model | Token Avg. | Inference Time(s) | GPU. util. | Memory (+15.87GB) |
|---|---|---|---|---|
| *1 × A6000 GPU Performance on MME* | | | | |
| Qwen2.5-VL-7B | 867.6 | 675 | 77.0% | 3.05 |
| VisionZip | 86.8 | 508 | 66.3% | 0.41 |
| DivPrune | 86.8 | 423 | 55.1% | 0.71 |
| MMTok | 86.8 | 419 | 60.0% | 0.71 |

Table 12: **Comparison of Inference Efficiency on Qwen2.5-VL-7B**. The initial memory usage for loading the model is 15.87GB.

### B.5 EFFICIENCY EFFECT OF THE NUMBER OF INPUT OR SELECTED VISION TOKENS

In MMTok, the similarity matrix can be constructed efficiently using the pytorch built-in libraries. Then, for token selection, MMTok proposes a greedy algorithm only with max operations and can run in $O(kn)$ to pick $k$ vision tokens, where $n$ is the total number of vision tokens. Therefore, our method can scale well with vision tokens and with the number of selected vision tokens. In Table 13, we show the running time of MMTok with the varying number of input and selected vision tokens. We find that with the same number of input tokens, the running time is almost linear in the number of selected tokens, which confirms our analysis. Moreover, even with 2880 input tokens, the running time of MMTok is less than 7ms, which is negligible for real applications. It should also be noted that even for 2880 input tokens, the computation only costs about 13.93 GFLOPs.

| #Input | #Select | Time(ms) | #Input | #Select | Time(ms) | #Input | #Select | Time(ms) |
|---|---|---|---|---|---|---|---|---|
| 2880 | 160 | 6.417 | 1728 | 96 | 3.862 | 576 | 32 | 1.267 |
| 2880 | 80 | 3.733 | 1728 | 48 | 2.247 | 576 | 16 | 0.774 |

Table 13: Running time (ms) of MMTok with different numbers of input and selected vision tokens on LLaVA-NeXT-7B. The reported result is averaged over 100 runs on a A6000 GPU.

### B.6 TOKEN SELECTION IN DECODER

Following the common practice, token selection has been conducted mainly after the vision encoder. In fact, token selection can also happen in the decoder. We conduct a preliminary experiment on the decoder in Table 14. We try to further select the vision tokens from 160 to 80 for an intermediate layer (i.e., L-24) of the decoder in LLaVA-Next on MME. We can find that it can keep the similar performance and further improve the token efficiency.

| Model | Upper Tokens | MME (P+C) ↑ |
|---|---|---|
| LLaVA-Next-13B | 2880 | 1901 |
| MMTok | 160 | 1811 |
| MMTok | 160⇒80 (L24) | 1846 |
| MMTok | 80 | 1717 |

Table 14: Comparison with vision token selection during decoding. We have an additional vision token selection in the 24th layer of decoder.

## C IMPROVED MMTOK

Since LLaVA-1.5 does not fine-tune the vision tower and also does not mask padding patches, we explicitly exclude padding patches from the candidate token set and fix an overflow bug that wasted one token. As shown in Table 15, these changes substantially improve accuracy while using fewer tokens. For a fair comparison, we report the results without the fix in the main text.

| Method | GQA Acc. ↑ | MMB Acc. ↑ | MME P+C ↑ | POPE F1 ↑ | SEED-I Acc. ↑ | Avg ↑ |
|---|---|---|---|---|---|---|
| *Vanilla Baseline (576 tokens)* | | | | | | |
| LLaVA-1.5-7B | 61.9 | 64.7 | 1862 | 85.9 | 66.14 | 100.0% |
| *32 Tokens* | | | | | | |
| MMTok | 55.95 | 58.59 | 1625 | 82.95 | 59.81 | 91.0% |
| MMTok++ | 56.61 | 58.76 | 1636 | 83.44 | 59.85 | 91.6% |
| *16 Tokens* | | | | | | |
| MMTok | 53.31 | 54.30 | 1551 | 79.79 | 56.67 | 86.4% |
| MMTok++ | 54.05 | 54.98 | 1581 | 80.79 | 57.13 | 87.5% |
| *8 Tokens* | | | | | | |
| MMTok | 49.06 | 49.06 | 1355 | 78.46 | 52.74 | 79.8% |
| MMTok++ | 50.80 | 49.31 | 1395 | 79.75 | 53.59 | 81.4% |
| *4 Tokens* | | | | | | |
| MMTok | 43.93 | 36.94 | 1290 | 74.84 | 48.10 | 71.4% |
| MMTok++ | 45.08 | 40.21 | 1294 | 76.36 | 49.34 | 73.6% |
| *2 Tokens* | | | | | | |
| MMTok | 40.58 | 25.69 | 1122 | 68.95 | 42.89 | 62.1% |
| MMTok++ | 42.18 | 31.36 | 1237 | 72.97 | 45.27 | 67.3% |
| *0 Tokens* | | | | | | |
| Baseline | 37.65 | 19.33 | 971 | 44.64 | 37.03 | 50.2% |

Table 15: Evaluate MMTok++ on LLaVA-1.5-7B with Extremely Less Token Budgets.

## D    SELECTED TOKENS VISUALIZATION

To provide an intuitive understanding of our token selection process, we visualize the selected tokens and their nearest words in Figure 5 and compare them with the diversity-based method, DivPrune. From the columns (b) and (c), we can observe that MMTok selects top patches according to the word to patch similarity, which aligns well with the question semantically. In contrast, as shown in columns (d) and (e), DivPrune selected top patches without any close semantic relation to the question. This further demonstrates that MMTok can help significantly reduce the number of tokens without losing the semantic relation to the questions, so as to provide better performance.

## E    COMPLETE EMPIRICAL RESULTS

This section shows per-dataset results for all models and token budgets, including LLaVA-1.5 (7B/13B) (Tables 16 and 17), LLaVA-NeXT (7B/13B) (Tables 18 and 19), and Qwen-2.5-VL (7B Table 20). We report both raw scores and percentage retention relative to the full-token setting. We also report results with an extremely low number of tokens on LLaVA-1.5-7B and LLaVA-NeXT-7B (Tables 21 and 22).

| Method | GQA Acc. ↑ | MMB Acc. ↑ | MME P+C ↑ | POPE F1 ↑ | SQA Acc. ↑ | VQA$^{V2}$ Acc. ↑ | VQA$^{Text}$ Acc. ↑ | MMMU Acc. ↑ | SEED Acc. ↑ | Avg. ↑ |
|---|---|---|---|---|---|---|---|---|---|---|
| *Upper Bound, 576 Tokens* **(100%)** | | | | | | | | | | |
| LLaVA-1.5 Vanilla 7B | 61.9 100% | 64.7 100% | 1862 100% | 85.9 100% | 69.5 100% | 78.5 100% | 58.2 100% | 36.3 100% | 58.6 100% | 100% |
| *Retain 192 Tokens* ↓ 66.7% | | | | | | | | | | |
| FastV (2024a) | 52.7 85.1% | 61.2 94.6% | 1612 86.6% | 64.8 75.4% | 67.3 96.8% | 67.1 85.5% | 52.5 90.2% | 34.3 94.5% | 57.1 97.4% | 89.6% |
| SparseVLM (2024) | 57.6 93.1% | 62.5 96.6% | 1721 92.4% | 83.6 97.3% | 69.1 99.4% | 75.6 96.3% | 56.1 96.4% | 33.8 93.1% | 55.8 95.2% | 95.5% |
| VisionZip (2025a) | 59.3 95.8% | 63.0 97.4% | 1782.6 95.7% | 85.3 99.3% | 68.9 99.1% | 76.8 97.8% | 57.3 98.5% | 36.6 100.8% | 56.4 96.2% | 97.9% |
| DivPrune (2025) | 59.97 96.9% | 62.54 96.7% | 1762.23 94.6% | 87.00 101.3% | 68.66 98.8% | 76.87 97.9% | 56.97 97.9% | 35.44 97.6% | 58.71 100.2% | 98.0% |
| VisionZip 🔥 (2025a) | 60.1 97.1% | 63.4 98.0% | 1834 98.5% | 84.9 98.8% | 68.2 98.1% | 77.4 98.6% | 57.8 99.3% | 36.2 99.7% | 57.1 97.4% | 98.4% |
| MMTok (Ours) | 60.07 97.0% | 63.40 98.0% | 1773.86 95.3% | 86.42 100.6% | 68.76 98.9% | 77.11 98.2% | 57.68 99.1% | 36.33 100.1% | 59.21 101.0% | **98.7%** |
| *Retain 128 Tokens* ↓ 77.8% | | | | | | | | | | |
| FastV (2024a) | 49.6 80.1% | 56.1 86.7% | 1490 80.0% | 59.6 69.4% | 60.2 86.6% | 61.8 78.7% | 50.6 86.9% | 34.9 96.1% | 55.9 95.4% | 84.4% |
| SparseVLM (2024) | 56.0 90.5% | 60.0 92.7% | 1696 91.1% | 80.5 93.7% | 67.1 96.5% | 73.8 94.0% | 54.9 94.3% | 33.8 93.1% | 53.4 91.1% | 92.9% |
| VisionZip (2025a) | 57.6 93.1% | 62.0 95.8% | 1761.7 94.6% | 83.2 96.9% | 68.9 99.1% | 75.6 96.3% | 56.8 97.6% | 37.9 104.4% | 54.9 93.7% | 96.8% |
| DivPrune (2025) | 59.25 95.7% | 62.03 95.9% | 1718.22 92.3% | 86.72 101.0% | 68.66 98.8% | 75.96 96.8% | 56.06 96.3% | 35.56 98.0% | 56.98 97.3% | 96.9% |
| VisionZip 🔥 (2025a) | 58.9 95.2% | 62.6 96.8% | 1823 97.9% | 83.7 97.4% | 68.3 98.3% | 76.6 97.6% | 57.0 97.9% | 37.3 102.8% | 55.8 95.2% | 97.7% |
| MMTok (Ours) | 59.29 95.8% | 62.29 96.3% | 1779.14 95.5% | 86.25 100.4% | 68.82 99.0% | 76.35 97.3% | 57.03 98.0% | 35.67 98.3% | 58.59 100.0% | **97.8%** |
| *Retain 64 Tokens* ↓ 88.9% | | | | | | | | | | |
| FastV (2024a) | 46.1 74.5% | 48.0 74.2% | 1256 67.5% | 48.0 55.9% | 51.1 73.5% | 55.0 70.1% | 47.8 82.1% | 34.0 93.7% | 51.9 88.6% | 75.6% |
| SparseVLM (2024) | 52.7 85.1% | 56.2 86.9% | 1505 80.8% | 75.1 87.4% | 62.2 89.4% | 68.2 86.9% | 51.8 89.0% | 32.7 90.1% | 51.1 87.2% | 86.9% |
| VisionZip (2025a) | 55.1 89.0% | 60.1 92.9% | 1690 90.8% | 77.0 89.6% | 69.0 99.3% | 72.4 92.2% | 55.5 95.4% | 36.2 99.7% | 52.2 89.1% | 93.2% |
| DivPrune (2025) | 57.78 93.3% | 59.28 91.6% | 1674.4 89.9% | 85.56 99.6% | 68.07 97.9% | 74.11 94.4% | 54.69 94.0% | 35.56 98.0% | 55.13 94.1% | 94.8% |
| VisionZip 🔥 (2025a) | 57.0 92.1% | 61.5 95.1% | 1756 94.3% | 80.9 94.2% | 68.8 99.0% | 74.2 94.5% | 56.0 96.2% | 35.6 98.1% | 53.4 91.1% | 95.0% |
| MMTok (Ours) | 58.29 94.2% | 61.17 94.5% | 1715.33 92.1% | 85.77 99.9% | 69.16 99.5% | 75.20 95.8% | 56.01 96.3% | 36.11 99.5% | 57.15 97.5% | **96.6%** |

Table 16: **Performance Comparison on LLaVA-1.5-7B.** The vanilla number of visual tokens is 576. The first line of each method shows the raw benchmark accuracy, and the second line is the proportion relative to the upper limit. The last column is the average value.

| Method | GQA Acc. ↑ | MMB Acc. ↑ | MME P+C ↑ | POPE F1 ↑ | SQA Acc. ↑ | VQA$^{V2}$ Acc. ↑ | VQA$^{Text}$ Acc. ↑ | MMMU Acc. ↑ | SEED-I Acc. ↑ | SEED$^*$ Acc. ↑ | Avg. ↑ |
|---|---|---|---|---|---|---|---|---|---|---|---|
| *Upper Bound, 576 Tokens* **(100%)** | | | | | | | | | | | |
| LLaVA-1.5 Vanilla 13B | 63.2 100% | 67.7 100% | 1818 100% | 85.9 100% | 72.8 100% | 80.0 100% | 61.3 100% | 36.4 100% | 66.9 100% | 61.6 100% | 100% |
| *Retain 192 Tokens* ↓ 66.7% | | | | | | | | | | | |
| VisionZip (2025a) | 59.1 93.5% | 66.9 98.8% | 1754 96.5% | 85.1 99.1% | 73.5 101.0% | 78.1 97.6% | 59.5 97.1% | 36.4 100% | 65.2 97.5% | 61.20† 99.4% | 97.9% |
| DivPrune (2025) | 59.42 94.0% | 66.58 98.3% | 1781.50 98.0% | 86.76 101.0% | 73.03 100.3% | 77.98 97.5% | 58.46 95.4% | 36.56 100.4% | 65.72 98.2% | 60.83 98.8% | 98.2% |
| VisionZip🔥 (2025a) | 61.6 97.5% | 67.1 99.1% | 1790 98.5% | 84.5 98.4% | 72.7 99.9% | 78.6 98.3% | 59.9 97.7% | 36.4 100% | 66.1 98.8% | – – | 98.7% |
| MMTok (Ours) | 59.67 94.4% | 67.70 100.0% | 1784.16 98.1% | 86.15 100.3% | 73.62 101.1% | 78.30 97.9% | 59.64 97.3% | 36.78 101.0% | 65.49 97.9% | 61.17 99.3% | **98.7%** |
| *Retain 128 Tokens* ↓ 77.8% | | | | | | | | | | | |
| VisionZip (2025a) | 57.9 91.6% | 66.7 98.5% | 1743 95.9% | 85.2↓ 99.2% | 74.0 101.6% | 76.8 96.0% | 58.7 95.8% | 36.1 99.2% | 63.8 95.4% | 59.74† 97.0% | 97.0% |
| DivPrune (2025) | 58.89 93.2% | 66.07 97.6% | 1748.56 96.2% | 86.53 100.7% | 72.48 99.6% | 77.10 96.4% | 58.17 94.9% | 35.56 97.7% | 64.22 96.0% | 59.49 96.6% | 96.9% |
| VisionZip🔥 (2025a) | 60.1 95.1% | 67.6 99.9% | 1736 95.5% | 83.8 97.6% | 73.0 100.3% | 77.6 97.0% | 59.2 96.6% | 35.4 97.3% | 64.9 97.0% | – – | 97.4% |
| MMTok (Ours) | 58.98 93.3% | 67.18 99.2% | 1756.20 96.6% | 86.22 100.4% | 73.38 100.8% | 77.57 97.0% | 59.22 96.6% | 35.44 97.4% | 64.26 96.1% | 60.11 97.6% | **97.5%** |
| *Retain 64 Tokens* ↓ 88.9% | | | | | | | | | | | |
| VisionZip (2025a) | 56.2 88.9% | 64.9 95.9% | 1676 92.2% | 76.0 88.5% | 74.4 102.2% | 73.7 92.1% | 57.4 93.6% | 36.4 100% | 60.4 90.3% | 57.13† 92.7% | 93.7% |
| DivPrune (2025) | 57.66 91.2% | 64.60 95.4% | 1777.93 97.8% | 84.80 98.7% | 72.09 99.0% | 75.20 94.0% | 57.11 93.2% | 35.22 96.8% | 62.44 93.3% | 57.70 93.7% | 95.3% |
| VisionZip🔥 (2025a) | 58.1 91.9% | 65.6 96.9% | 1671 91.9% | 81.6 95.0% | 72.3 99.3% | 75.2 94.0% | 58.5 95.4% | 35.3 97.0% | 61.4 91.8% | – – | 94.8% |
| MMTok (Ours) | 58.42 92.4% | 65.72 97.1% | 1763.39 97.0% | 84.39 98.2% | 72.98 100.2% | 76.55 95.7% | 58.40 95.3% | 35.22 96.8% | 63.39 94.8% | 59.51 96.6% | **96.4%** |

Table 17: **Performance Comparison on LLaVA-1.5-13B.** The vanilla number of visual tokens is 576. The first line of each method shows the raw benchmark accuracy, and the second line is the proportion relative to the upper limit. SEED-I represents SEED-IMG, SEED represents SEED-ALL. Following (Yang et al., 2025a), Avg. is based on SEED-I instead of SEED.

| Method | GQA Acc. ↑ | MMB Acc. ↑ | MME P+C ↑ | POPE F1 ↑ | SQA Acc. ↑ | VQA$^{V2}$ Acc. ↑ | VQA$^{Text}$ Acc. ↑ | MMMU Acc. ↑ | SEED-I Acc. ↑ | Avg. ↑ |
|---|---|---|---|---|---|---|---|---|---|---|
| Avg. Images $\bar{n}$ | 4.90 | 4.12 | 4.53 | 4.90 | 3.85 | 4.98 | 4.98 | 4.07 | 4.72 | |
| Avg. Tokens ($\bar{n} * 576$) | 2822.4 | 2373.12 | 2609.28 | 2822.40 | 2217.60 | 2868.48 | 2868.48 | 2344.32 | 2718.72 | |
| *Upper Bound: 2880 (5 × 576) Tokens* **(100%)** | | | | | | | | | | |
| LLaVA-NeXT Vanilla 7B | 64.2 100% | 67.9 100% | 1842 100% | 86.4 100% | 70.2 100% | 80.1 100% | 61.3 100% | 35.1 100% | 70.2 100% | 100% |
| *Upper: 5 × 128 = 640* | 627 | 527 | 580 | 627 | 493 | 638 | 638 | 521 | 604 | ↓ **77.8%** |
| SparseVLM (2024) | 60.3 93.9% | 65.7 96.8% | 1772 96.2% | – – | 67.7 96.4% | 77.1 96.3% | 57.8 94.3% | 34.6 98.6% | – – | - |
| VisionZip (2025a) | 61.3 95.5% | 66.3 97.6% | 1787 97.0% | 86.3 99.9% | 68.1 97.0% | 79.1 98.8% | 60.2 98.2% | 34.7 98.9% | 66.7 95.0% | 97.5% |
| DivPrune (2025) | 61.58 95.9% | 65.38 96.3% | 1773.04 96.3% | 85.51 99.0% | 67.82 96.6% | 78.94 98.6% | 55.41 90.4% | 36.89 105.1% | 67.56 96.2% | 97.1% |
| VisionZip ♦ (2025a) | 62.4 97.2% | 65.9 97.1% | 1778 96.5% | 87.6 101.4% | 67.9 96.7% | 79.9 99.8% | 60.8 99.2% | 37.2 106.0% | 67.8 96.6% | 98.9% |
| MMTok (Ours) | 62.27 97.0% | 65.29 96.2% | 1829.28 99.3% | 86.74 100.4% | 68.47 97.5% | 79.31 99.0% | 58.97 96.2% | 37.22 106.0% | 67.74 96.5% | **98.7%** |
| *Upper: 5 × 64 = 320* | 314 | 264 | 290 | 314 | 246 | 319 | 319 | 261 | 302 | ↓ **88.9%** |
| SparseVLM (2024) | 57.7 89.9% | 64.3 94.7% | 1694 92.0% | – – | 67.3 95.9% | 73.4 91.6% | 55.9 91.2% | 34.4 98.0% | – – | - |
| VisionZip (2025a) | 59.3 92.4% | 63.1 92.9% | 1702 92.4% | 82.1 95.0% | 67.3 95.9% | 76.2 95.1% | 58.9 96.1% | 35.3 100.6% | 63.4 90.3% | 94.5% |
| DivPrune (2025) | 59.63 92.9% | 63.66 93.7% | 1731.04 94.0% | 83.47 96.6% | 67.82 96.6% | 76.64 95.7% | 53.84 87.8% | 37.11 105.7% | 65.35 93.1% | 95.1% |
| VisionZip ♦ (2025a) | 61.0 95.0% | 64.4 94.8% | 1770 96.1% | 86.2 99.8% | 67.5 96.2% | 78.4 97.9% | 59.3 96.7% | 38.0 108.3% | 65.9 93.9% | 97.6% |
| MMTok (Ours) | 60.96 95.0% | 64.35 94.8% | 1799.33 97.7% | 85.76 99.3% | 67.33 95.9% | 77.68 97.0% | 56.93 92.9% | 38.00 108.3% | 66.29 94.4% | **97.3%** |
| *Upper: 5 × 32 = 160* | 157 | 132 | 145 | 157 | 123 | 159 | 159 | 130 | 151 | ↓ **94.4%** |
| SparseVLM (2024) | 51.2 79.8% | 63.1 92.9% | 1542 83.7% | – – | 67.5 96.2% | 66.3 82.8% | 46.4 75.7% | 32.8 93.4% | – – | - |
| VisionZip (2025a) | 55.5 86.4% | 60.1 88.5% | 1630 88.5% | 74.8 86.6% | 68.3 97.3% | 71.4 89.1% | 56.2 91.7% | 36.1 102.8% | 58.3 83.0% | 90.4% |
| DivPrune (2025) | 57.79 90.0% | 62.29 91.7% | 1658.25 90.0% | 79.36 91.9% | 68.02 96.9% | 73.92 92.3% | 52.42 85.5% | 36.44 103.8% | 62.54 89.1% | 92.4% |
| VisionZip ♦ (2025a) | 58.2 90.7% | 63.9 94.1% | 1699 92.2% | 83.4 96.5% | 67.5 96.2% | 75.6 94.4% | 57.3 93.5% | 37.7 107.4% | 62.9 89.6% | 95.0% |
| MMTok (Ours) | 60.05 93.5% | 62.97 92.7% | 1715.54 93.1% | 83.87 97.1% | 67.97 96.8% | 75.62 94.4% | 54.17 88.4% | 37.89 107.9% | 64.54 91.9% | **95.1%** |

Table 18: **Performance Comparison on LLaVA-NeXT-7B.** The vanilla number of visual tokens varies by dataset due to dynamic image processing (max 2880 for 5 images). '-' means performance not available in the original paper.

| Method | GQA Acc. ↑ | MMB Acc. ↑ | MME P+C ↑ | POPE F1 ↑ | SQA Acc. ↑ | VQA$^{V2}$ Acc. ↑ | VQA$^{Text}$ Acc. ↑ | MMMU Acc. ↑ | SEED-I Acc. ↑ | Avg. ↑ |
|---|---|---|---|---|---|---|---|---|---|---|
| Avg. Images $\bar{n}$ | 4.90 | 4.12 | 4.53 | 4.90 | 3.85 | 4.98 | 4.98 | 4.07 | 4.72 | |
| Avg. Tokens ($\bar{n} * 576$) | 2822.4 | 2373.12 | 2609.28 | 2822.40 | 2217.60 | 2868.48 | 2868.48 | 2344.32 | 2718.72 | |
| *Upper Bound: 2880 (5 × 576) Tokens* **(100%)** | | | | | | | | | | |
| LLaVA-NeXT Vanilla 13B | 65.4 | 70.0 | 1901 | 86.2 | 73.5 | 81.8 | 64.3 | 36.2 | 71.9 | 100% |
| | 100% | 100% | 100% | 100% | 100% | 100% | 100% | 100% | 100% | |
| *Upper: 5 × 128 = 640* | 627 | 527 | 580 | 627 | 493 | 638 | 638 | 521 | 604 | ↓**77.8%** |
| VisionZip (2025a) | 63.0 | 68.6 | 1871 | 85.7 | 71.2 | 79.7 | 62.2 | 36.4 | 68.8 | 97.7% |
| | 96.3% | 98.0% | 98.4% | 99.4% | 96.9% | 97.4% | 96.7% | 100.5% | 95.7% | |
| DivPrune (2025) | 62.82 | 66.84 | 1832.76 | 86.17 | 71.84 | 79.87 | 57.54 | 37.78 | 69.38 | 97.1% |
| | 96.1% | 95.5% | 96.4% | 99.9% | 97.7% | 97.6% | 89.5% | 104.4% | 96.5% | |
| VisionZip ♦ (2025a) | 63.7 | 66.6 | 1829 | 86.3 | 73.2 | 81.2 | 64.4 | 38.1 | 69.2 | 98.8% |
| | 97.4% | 95.1% | 96.2% | 100.1% | 99.6% | 99.3% | 100.2% | 105.2% | 96.2% | |
| MMTok (Ours) | 63.71 | 67.44 | 1874.63 | 86.72 | 72.29 | 80.55 | 61.06 | 37.11 | 69.61 | **98.2%** |
| | 97.4% | 96.3% | 98.6% | 100.6% | 98.4% | 98.5% | 95.0% | 102.5% | 96.8% | |
| *Upper: 5 × 64 = 320* | 314 | 264 | 290 | 314 | 246 | 319 | 319 | 261 | 302 | ↓**88.9%** |
| VisionZip (2025a) | 60.7 | 67.2 | 1805 | 82.0 | 70.3 | 76.8 | 60.9 | 35.6 | 65.2 | 94.7% |
| | 92.8% | 96.0% | 95.0% | 95.1% | 95.6% | 93.9% | 94.7% | 98.3% | 90.7% | |
| DivPrune (2025) | 61.03 | 65.46 | 1802.79 | 84.86 | 71.39 | 77.6 | 55.75 | 36.00 | 66.75 | 94.5% |
| | 93.3% | 93.5% | 94.8% | 98.4% | 97.1% | 94.9% | 86.7% | 99.4% | 92.8% | |
| VisionZip ♦ (2025a) | 62.5 | 66.9 | 1861 | 85.7 | 72.7 | 80.0 | 63.2 | 36.9 | 67.9 | 97.8% |
| | 95.6% | 95.6% | 97.9% | 99.4% | 98.9% | 97.8% | 98.3% | 101.9% | 94.4% | |
| MMTok (Ours) | 62.95 | 65.55 | 1840.10 | 85.88 | 72.38 | 78.79 | 58.88 | 36.33 | 67.81 | **96.4%** |
| | 96.3% | 93.6% | 96.8% | 99.6% | 98.5% | 96.3% | 91.6% | 100.4% | 94.3% | |
| *Upper: 5 × 32 = 160* | 157 | 132 | 145 | 157 | 123 | 159 | 159 | 130 | 151 | ↓**94.4%** |
| VisionZip (2025a) | 57.8 | 64.9 | 1739 | 76.6 | 69.3 | 72.4 | 58.4 | 37.0 | 61.1 | 91.4% |
| | 88.4% | 92.7% | 91.5% | 88.9% | 94.3% | 88.5% | 90.8% | 102.2% | 85.0% | |
| DivPrune (2025) | 59.34 | 64.78 | 1699.83 | 82.16 | 70.55 | 74.72 | 54.65 | 35.89 | 63.80 | 92.0% |
| | 90.7% | 92.5% | 89.4% | 95.3% | 96.0% | 91.3% | 85.0% | 99.1% | 88.7% | |
| VisionZip ♦ (2025a) | 59.7 | 65.3 | 1766 | 84.0 | 72.0 | 77.6 | 60.8 | 36.0 | 64.4 | 94.6% |
| | 91.3% | 93.3% | 92.9% | 97.4% | 98.0% | 94.9% | 94.6% | 99.4% | 89.6% | |
| MMTok (Ours) | 61.94 | 65.89 | 1811.35 | 85.11 | 72.43 | 76.8 | 55.91 | 37.11 | 65.45 | **95.1%** |
| | 94.7% | 94.1% | 95.3% | 98.7% | 98.5% | 93.9% | 87.0% | 102.5% | 91.0% | |

Table 19: **Performance Comparison on LLaVA-NeXT-13B.** The vanilla upper number of visual tokens is 2880. SEED-I represents SEED-IMG.

| Method | GQA Acc. ↑ | MMB Acc. ↑ | MME P+C ↑ | POPE F1 ↑ | VQA$^{\text{Text}}$ Acc. ↑ | SQA Acc. ↑ | OCRBench Acc. ↑ | Avg.† ↑ |
|---|---|---|---|---|---|---|---|---|
| *Dynamic Resolution (MinPix = 256 × 28 × 28, MaxPix = 2048 × 28 × 28), Upper Bound* **(100%)** | | | | | | | | |
| Avg. Tokens $\bar{T}$ | 358.5 | 276.9 | 867.6 | 359.6 | 976.5 | 323.0 | 652.8 | |
| Qwen-2.5-VL-7B Dynamic Res. | 60.48 100% | 83.25 100% | 2327 100% | 86.16 100% | 77.72 100% | 87.46 100% | 83.80 100% | 100% |
| *Fixed Resolution (MinPix = MaxPix = 2048 × 28 × 28), Upper Bound* **(100%)** | | | | | | | | |
| Qwen-2.5-VL-7B Fixed Res. | 58.59 96.9% | 83.59 100.4% | 2339 100.5% | 86.09 99.9% | 76.64 98.6% | 86.91 99.4% | 76.60 91.4% | 99.3% |
| *Retain 20% $\bar{T}$* | 71.7 | 55.4 | 173.5 | 71.9 | 195.3 | 64.6 | 130.6 | ↓ **80%** |
| VisionZip (2025a) | 56.80 93.9% | 80.33 96.5% | 2174 93.4% | 83.38 96.8% | 70.43 90.6% | 84.23 96.3% | 59.50 71.0% | 94.2% |
| DivPrune (2025) | 56.70 93.8% | 76.98 92.5% | 2163 93.0% | 80.59 93.5% | 65.86 84.7% | 80.91 92.5% | 48.10 57.4% | 91.5% |
| MMTok (Ours) | 58.09 96.0% | 79.30 95.3% | 2217 95.3% | 82.38 95.7% | 70.49 90.7% | 81.61 93.3% | 59.60 71.1% | **94.6%** |
| *Retain 10% $\bar{T}$* | 35.9 | 27.7 | 86.8 | 36.0 | 97.7 | 32.3 | 65.3 | ↓ **90%** |
| VisionZip (2025a) | 52.47 86.8% | 75.60 90.8% | 2003 86.1% | 78.90 91.6% | 63.78 82.1% | 82.30 94.1% | 36.90 44.0% | 87.5% |
| DivPrune (2025) | 53.43 88.3% | 72.85 87.5% | 1957 84.1% | 74.99 87.0% | 59.59 76.7% | 79.57 91.0% | 37.30 44.5% | 84.7% |
| MMTok (Ours) | 55.09 91.1% | 74.74 89.8% | 2051 88.1% | 78.75 91.4% | 63.90 82.2% | 80.47 92.0% | 43.60 52.1% | **88.5%** |
| *Retain 5% $\bar{T}$* | 17.9 | 13.8 | 43.4 | 18.0 | 48.8 | 16.2 | 32.6 | ↓ **95%** |
| VisionZip (2025a) | 46.28 76.5% | 67.53 81.1% | 1677 72.1% | 66.38 77.1% | 54.49 70.1% | 79.57 91.0% | 19.70 23.5% | 75.4% |
| DivPrune (2025) | 49.01 81.0% | 65.89 79.1% | 1739 74.7% | 68.45 79.4% | 52.02 66.9% | 77.05 88.1% | 24.90 29.7% | 76.3% |
| MMTok (Ours) | 50.66 83.8% | 65.89 79.2% | 1796 77.2% | 71.35 82.8% | 55.95 72.0% | 77.19 88.2% | 30.70 36.6% | **79.0%** |
| *0 Token* ↓ **100%** | | | | | | | | |
| Qwen-2.5-VL 7B Text-Only | 31.84 54.3% | 20.10 24.0% | 935 40.0% | 0.00* 0.0%* | 38.93 50.8% | 71.10 88.6% | 1.80 2.1% | 33.8% |

Table 20: **Performance Comparison on Qwen-2.5-VL-7B-Instruct.** Avg.† is the average performance over the 5 datasets: GQA, MMB, MME, POPE, and VQA$^{\text{Text}}$. The first line of each method shows the raw benchmark accuracy, and the second line is the proportion relative to the upper limit. *Qwen-2.5-VL outputs `No` for all POPE questions when no visual tokens are provided, resulting in 0% F1 score.

| Method | GQA Acc. ↑ | MMB Acc. ↑ | MME P+C ↑ | POPE F1 ↑ | SQA Acc. ↑ | VQA$^{\text{Text}}$ Acc. ↑ | MMMU Acc. ↑ | SEED-I* Acc. ↑ | Avg@8 ↑ | Avg@5 ↑ | ≥90% /8 ↑ | ≥80% /8 ↑ |
|---|---|---|---|---|---|---|---|---|---|---|---|---|
| *Vanilla Baseline (576 tokens)* | | | | | | | | | | | | |
| LLaVA-1.5-7B | 61.9 | 64.7 | 1862 | 85.9 | 69.5 | 58.2 | 36.3 | 66.14 | 100.0% | 100.0% | 8/8 | 8/8 |
| 90% Threshold | 55.71 | 58.23 | 1675.80 | 77.31 | 62.55 | 52.38 | 32.67 | 59.53 | 90.0% | 90.0% | – | – |
| 80% Threshold | 49.52 | 51.76 | 1489.60 | 68.72 | 55.60 | 46.56 | 29.04 | 52.91 | 80.0% | 80.0% | – | – |
| *64 Tokens* | | | | | | | | | | | | |
| VisionZip | 55.1 | 60.1 | 1690 | 77.0 | 69.0 | 55.5 | 36.2 | 57.84 | 93.0% | 90.0% | 5/8 | 8/8 |
| DivPrune | 57.78 | 59.28 | 1674.40 | 85.56 | 68.07 | 54.69 | 35.56 | 60.21 | 94.4% | 93.1% | 7/8 | 8/8 |
| MMTok | 58.29 | 61.17 | 1715.33 | 85.77 | 69.16 | 56.01 | 36.11 | 61.29 | 96.1% | 94.7% | 8/8 | 8/8 |
| *32 Tokens* | | | | | | | | | | | | |
| VisionZip | 51.78 | 57.22 | 1580.43 | 68.88 | 68.77 | 53.23 | 35.11 | 53.28 | 88.1% | 83.5% | 3/8 | 8/8 |
| DivPrune | 55.11 | 58.93 | 1600 | 82.06 | 68.62 | 53.20 | 35.33 | 57.08 | 91.9% | 89.6% | 5/8 | 8/8 |
| MMTok | 55.95 | 58.59 | 1624.72 | 82.95 | 68.86 | 53.70 | 35.33 | 59.81 | 93.0% | 91.0% | 7/8 | 8/8 |
| *16 Tokens* | | | | | | | | | | | | |
| VisionZip | 46.72 | 45.70 | 1326.89 | 51.84 | 67.67 | 49.74 | 35.00 | 46.66 | 78.4% | 69.7% | 2/8 | 3/8 |
| DivPrune | 51.10 | 53.09 | 1518 | 69.56 | 69.41 | 50.01 | 35.44 | 52.72 | 86.3% | 81.4% | 2/8 | 7/8 |
| MMTok | 53.31 | 54.30 | 1550.65 | 79.79 | 68.82 | 50.04 | 34.22 | 56.67 | 88.9% | 86.4% | 3/8 | 8/8 |
| *8 Tokens* | | | | | | | | | | | | |
| VisionZip | 39.47 | 24.40 | 1069.94 | 23.66 | 64.30 | 44.62 | 33.67 | 38.46 | 63.3% | 48.9% | 2/8 | 2/8 |
| DivPrune | 46.09 | 43.13 | 1294 | 52.10 | 67.92 | 45.21 | 34.00 | 46.68 | 76.4% | 68.4% | 2/8 | 2/8 |
| MMTok | 49.06 | 49.06 | 1355.31 | 78.46 | 66.83 | 45.71 | 34.11 | 52.74 | 83.5% | 79.8% | 3/8 | 3/8 |
| *4 Tokens* | | | | | | | | | | | | |
| VisionZip | 36.57 | 18.30 | 923.57 | 24.48 | 63.56 | 40.82 | 33.78 | 35.34 | 59.2% | 43.8% | 2/8 | 2/8 |
| DivPrune | 40.67 | 28.61 | 1134 | 33.33 | 65.20 | 42.54 | 33.33 | 40.99 | 66.3% | 54.3% | 2/8 | 2/8 |
| MMTok | 43.93 | 36.94 | 1290.31 | 74.84 | 65.64 | 43.52 | 34.00 | 48.10 | 77.5% | 71.4% | 2/8 | 3/8 |
| *2 Tokens* | | | | | | | | | | | | |
| VisionZip | 35.94 | 16.84 | 890.28 | 26.48 | 63.31 | 39.55 | 33.78 | 34.62 | 58.4% | 43.0% | 2/8 | 2/8 |
| DivPrune | 38.58 | 21.48 | 991 | 37.60 | 64.60 | 42.16 | 33.44 | 38.43 | 63.5% | 50.1% | 2/8 | 2/8 |
| MMTok | 40.58 | 25.69 | 1122.42 | 68.95 | 64.90 | 42.42 | 32.67 | 42.89 | 70.9% | 62.1% | 2/8 | 3/8 |
| *0 Tokens* | | | | | | | | | | | | |
| Baseline | 37.65 | 19.33 | 970.89 | 44.64 | 63.51 | 41.66 | 33.33 | 37.03 | 63.2% | 50.2% | 2/8 | 2/8 |

Table 21: **Extended Performance Comparison with Extremely Less Token Budgets on LLaVA-1.5-7B.** *SEED-I indicts SEEDBench-Image. Avg@8 is across all 8 datasets, while Avg@5 is on 5 High-IC datasets.

| Method | GQA Acc. ↑ | MMB Acc. ↑ | MME P+C ↑ | POPE F1 ↑ | SQA Acc. ↑ | VQA^Text Acc. ↑ | MMMU Acc. ↑ | SEED-I Acc. ↑ | Avg@8 ↑ | Avg@6 ↑ | ≥90% /8 ↑ | ≥80% /8 ↑ |
|---|---|---|---|---|---|---|---|---|---|---|---|---|
| *Vanilla Baseline (Upper 2880 tokens)* | | | | | | | | | | | | |
| LLaVA-NeXT-7B | 64.2 | 67.9 | 1842 | 86.4 | 70.2 | 61.3 | 35.1 | 70.2 | 100.0% | 100.0% | 8/8 | 8/8 |
| 90% Threshold | 57.78 | 61.11 | 1657.80 | 77.76 | 63.18 | 55.17 | 31.59 | 63.18 | 90.0% | 90.0% | – | – |
| 80% Threshold | 51.36 | 54.32 | 1473.60 | 69.12 | 56.16 | 49.04 | 28.08 | 56.16 | 80.0% | 80.0% | – | – |
| *Upper 32×5 Tokens (160 Tokens) 5.6%* | | | | | | | | | | | | |
| VisionZip | 55.5 | 60.1 | 1630 | 74.8 | 68.3 | 56.2 | 36.1 | 58.3 | 90.6% | 87.5% | 3/8 | 8/8 |
| DivPrune | 57.79 | 62.29 | 1658 | 79.36 | 68.02 | 52.42 | 36.44 | 62.54 | 92.4% | 89.7% | 6/8 | 8/8 |
| MMTok | 60.05 | 62.97 | 1716 | 83.87 | 67.97 | 54.17 | 37.89 | 64.54 | 95.2% | 92.8% | 7/8 | 8/8 |
| *Upper 16×5 Tokens (80 Tokens) 2.8%* | | | | | | | | | | | | |
| VisionZip | 50.80 | 50.69 | 1431 | 61.82 | 66.93 | 51.65 | 34.44 | 51.77 | 81.8% | 76.8% | 2/8 | 3/8 |
| DivPrune | 55.73 | 59.97 | 1575 | 74.74 | 66.83 | 50.35 | 36.56 | 59.48 | 89.2% | 85.7% | 2/8 | 8/8 |
| MMTok | 58.23 | 62.54 | 1681 | 81.89 | 67.13 | 49.56 | 36.11 | 61.86 | 92.0% | 89.6% | 6/8 | 8/8 |
| *Upper 8×5 Tokens (40 Tokens) 1.4%* | | | | | | | | | | | | |
| VisionZip | 41.87 | 28.35 | 999 | 21.22 | 64.25 | 42.85 | 31.44 | 41.93 | 62.1% | 52.6% | 1/8 | 2/8 |
| DivPrune | 52.87 | 55.76 | 1462 | 67.49 | 66.78 | 48.02 | 33.44 | 55.40 | 83.7% | 79.9% | 2/8 | 4/8 |
| MMTok | 54.52 | 59.88 | 1555 | 81.84 | 67.73 | 45.77 | 35.00 | 59.28 | 88.4% | 85.2% | 3/8 | 7/8 |
| *Upper 4×5 Tokens (20 Tokens) 0.7%* | | | | | | | | | | | | |
| VisionZip | 36.56 | 18.38 | 814 | 0.40 | 63.56 | 35.36 | 31.56 | 34.98 | 52.1% | 39.4% | 1/8 | 2/8 |
| DivPrune | 49.57 | 48.54 | 1324 | 52.14 | 65.94 | 44.06 | 31.78 | 51.25 | 76.3% | 71.0% | 2/8 | 2/8 |
| MMTok | 49.60 | 51.20 | 1457 | 82.41 | 66.88 | 42.33 | 33.67 | 55.34 | 83.3% | 79.2% | 3/8 | 3/8 |
| *Upper 2×5 Tokens (10 Tokens) 0.3%* | | | | | | | | | | | | |
| VisionZip | 36.17 | 17.96 | 823 | 0.80 | 62.91 | 32.84 | 30.56 | 34.31 | 50.9% | 38.5% | 0/8 | 2/8 |
| DivPrune | 45.19 | 37.11 | 1134 | 25.48 | 65.25 | 40.33 | 33.22 | 45.54 | 66.8% | 57.8% | 2/8 | 2/8 |
| MMTok | 45.72 | 38.75 | 1283 | 79.62 | 65.64 | 39.77 | 33.78 | 49.73 | 76.9% | 71.0% | 3/8 | 3/8 |
| *0 Tokens* | | | | | | | | | | | | |
| Baseline | 38.23 | 17.87 | 867 | 25.84 | 64.60 | 37.77 | 31.56 | 37.43 | 57.5% | 46.3% | 1/8 | 2/8 |

Table 22: **Extended Performance Comparison with Extremely Less Token Budgets on LLaVA-NeXT-7B.** Avg@8 is across all 8 datasets, while Avg@6 is across 6 High-IC datasets. The "×5" notation indicates maximum sampling to 5 images. Average percentages are calculated relative to the vanilla baseline for each metric.

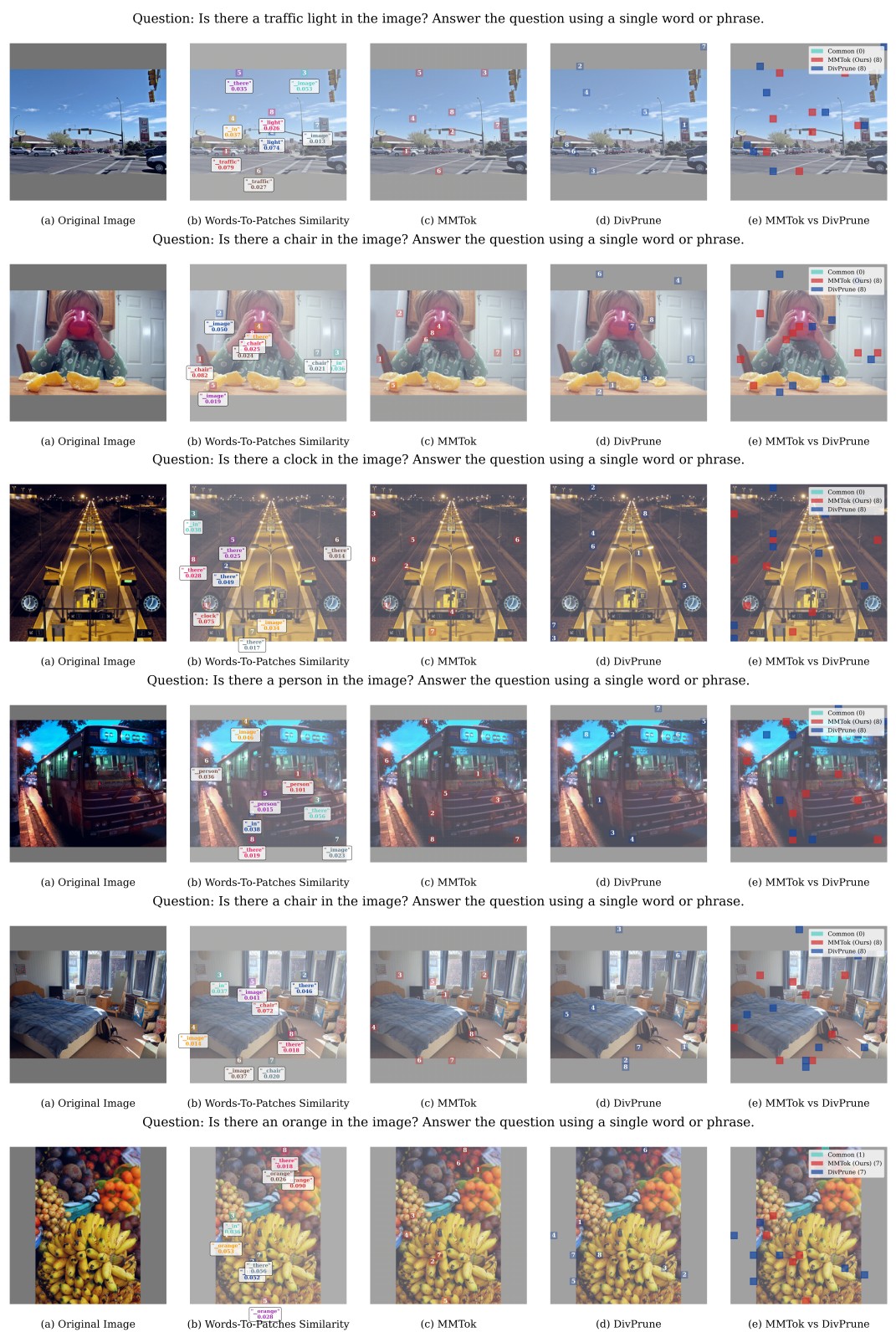

Figure 5: **Visualization of Selected Tokens.** Compared with the diversity-based method, DivPrune, our method selected token coverage the necessary token associate to language context.

