# OpenReview forum: "MMTok: Multimodal Coverage Maximization for Efficient Inference of VLMs"
_ICLR.cc/2026/Conference — ICLR 2026 Poster_

### Official Review · Reviewer_EEd7 · 2025-10-25

**Soundness:** 3
**Presentation:** 3
**Contribution:** 3
**Rating:** 6
**Confidence:** 3

**Summary:**

This paper proposes a novel token pruning method for VLMs. The authors formulate token pruning as a maximum coverage submodular optimization problem: finding a subset of visual tokens that covers both the textual tokens (text-visual coverage) and the visual tokens themselves (visual-visual coverage). The authors also provide an optional agent (MMTok-Agent) that enriches text guidance by adding preliminary text output from a small VLM. Experimental results show that the proposed method achieves significant speedup while pruning most tokens.

**Strengths:**

1. The paper is well-written, with easy-to-understand figures and tables.

2. The authors provide excellent theoretical support, supplemented by experiments. The design is well-designed.

3. The experiments are solid, experimented on nine datasets and various virtual machine models, demonstrating robust generalization capabilities.

**Weaknesses:**

1. The role of the agent is not very clear. From the experiments, it seems to bring only slightly improvements. Although the authors claim it does not affect inference efficiency, it still introduces additional computation, which somewhat contradicts the very movitation of token pruning.

2. The authors provide solid theoretical justification and a novel selection methods; however, this method appears to be applicable only to the encoder side. Why not consider expand it to decoder? Please correct me if my understanding is mistake.

**Questions:**

1. Unlike most approaches that select tokens at the decoder stage, what are the advantages of selecting tokens at the encoder stage? Why can't apply to both the encoder and decoder?

2. Are there any datasets where performance drops significantly at higher pruning rates, while others remain relatively stable?

3. To my limit knowledge, token pruning methods can ensure that overall performance does not drop significantly. However, they may introduce knowledge boundary drift, meaning that answers that were previously correct may now be incorrect, and vice versa. This possibility could have negative consequences in scenarios where critical questions must be answered correctly. If possible, could the authors conduct a simple analysis to see whether the proposed method leads to changes in individual case performance?

---

> ### Author Response · Authors · 2025-11-26
>
> We would like to thank the reviewer for valuable time and insightful comments. In the uploaded revision, we addressed all raised concerns as follows.
>
> Q1: The role of the agent is not very clear.
>
> A1: Thanks for pointing this out. Since the main contribution of this work is the maximum coverage for token selection while the agent is optional, we removed this option in the revision and leave the study of agent for vision token selection as our future work to avoid any misleading.
>
> Q2: This method appears to be applicable only to the encoder side. Why not consider expand it to decoder? Please correct me if my understanding is mistake.
>
> A2: Thanks for the great suggestion. We conduct the preliminary experiments on decoder as follows. We try to further prune the vision tokens from 160 to 80 for an intermediate layer (i.e., L-24) of the decoder in LLaVA-Next on MME. We can find that it can keep the similar performance and further improve the token efficiency. We have added the discussion for decoder in the revision's appendix B.7.
>
> Table: Comparison with vision token selection during decoding. We have an additional vision token selection in the 24th layer of decoder.
>
> | Model | Upper Tokens | MME P+G ↑ |
> |-------|--------------|-----------|
> | LLaVA-Next-13B | 2880 | 1901 |
> | Ours | 160 | 1811 |
> | Ours | 160→80 (L24) | 1846 |
> | Ours | 80 | 1717 |
>
> Q3: Unlike most approaches that select tokens at the decoder stage, what are the advantages of selecting tokens at the encoder stage? Why can't apply to both the encoder and decoder?
>
> A3: All approaches in the comparison select vision tokens after the vision encoder. Therefore, we follow their settings in the experiment for a fair comparison. However, selecting tokens at the decoder stage is possible and is a great suggestion as mentioned above. We have included the discussion in the revision's appendix B.7.
>
> Q4: Are there any datasets where performance drops significantly at higher pruning rates, while others remain relatively stable?
>
> A4: Table 16 in the original submission (now Table 21 in the revision) shows the detailed results with a high pruning ratio and we can find that our method still works well on POPE with only 4 vision tokens and the other baselines drop significantly.
>
> Q5: Could the authors conduct a simple analysis to see whether the proposed method leads to changes in individual case performance?
>
> A5: Since the retained performance is not 100\%, the performance can change in an individual case after token selection. In the revision, we added some examples in Figure 4. We find that the answer changes when the number of vision tokens is reduced significantly. Therefore, for critical questions, we may keep more vision tokens for consistency. In addition, a simple question will work well with less vision tokens as in the first example, which implies that the hardness of question is also important for answer drift. It is a very interesting direction and exploring this direction further can be our future work. Many thanks for your suggestion.

---

> > ### Comment · Reviewer_EEd7 · 2025-11-26
> >
> > Thank you for the comprehensive rebuttal. The clarifications provided effectively resolve my concerns. As my initial overall assessment was positive, I will maintain my score.

---

> > > ### Author Response · Authors · 2025-11-26
> > >
> > > We are grateful that your concerns have been well addressed. Your insightful comments have significantly improved this work and inspired our future work. Thank you again for your valuable feedback.

---

### Official Review · Reviewer_TSCU · 2025-10-27

**Soundness:** 3
**Presentation:** 3
**Contribution:** 2
**Rating:** 6
**Confidence:** 4

**Summary:**

This paper introduces MMTok, a multimodal token selection framework designed to enhance the inference efficiency of Vision-Language Models. The key idea is to leverage both vision and text modalities for vision token pruning, rather than relying on unimodal information as in prior work. The authors formulate the token selection as a maximum coverage problem, aiming to optimize a subset of vision tokens that best cover both text tokens and the original visual space. Additionally, a VLM agent is utilized to refine text representations, further improving pruning quality. Experimental results on multiple benchmark datasets and various VLM architectures demonstrate that MMTok effectively reduces redundancy and accelerates inference.

**Strengths:**

1. Formulating vision token pruning as a maximum coverage problem is both novel and insightful. This formulation naturally captures visual–textual interactions by optimizing vision tokens to jointly cover textual semantics and the original visual space, providing a solid mathematical foundation for multimodal token selection.

2. The authors conduct impressive experiments on five models and demonstrate improvements over previous methods. Moreover, the ablation study is detailed, particularly in the efficiency analysis section.

3. The visualizations presented in the Appendix are impressive and highly detailed.

**Weaknesses:**

1. The proposed VLM agent offers limited contribution. As shown in Table 1, it yields at most a 0.2 improvement and even leads to performance degradation in some cases, while introducing additional time overhead.

2. Compared with finetuned VisionZip, MMTok performs notably better under the 64-token setting. However, its advantage diminishes under the 128- and 192-token settings. I suggest evaluating MMTok on more challenging benchmarks, such as MMStar and MathVista, to better highlight its strengths.

**Questions:**

1. Is there a **fundamental difference** between Coverage Maximization and maximizing Conditional Diversity for Token Pruning? This is important, as there are already works that perform token pruning based on differentiable diversity (e.g., [1]).

[1] Beyond Attention or Similarity: Maximizing Conditional Diversity for Token Pruning in MLLMs, NeurIPS 2025.

---

> ### Author Response · Authors · 2025-11-26
>
> We would like to thank the reviewer for valuable time and insightful comments. In the uploaded revision, we addressed all raised concerns as follows.
>
> Q1: The proposed VLM agent offers limited contribution, while introducing additional time overhead.
>
> A1: Thanks for pointing this out. Since the main contribution of this work is the maximum coverage for token selection while the agent is optional, we removed this option in the revision and leave the study of agent for vision token selection as our future work to avoid any misleading.
>
> Q2: Compared with finetuned VisionZip, MMTok performs notably better under the 64-token setting. However, its advantage diminishes under the 128- and 192-token settings. I suggest evaluating MMTok on more challenging benchmarks, such as MMStar and MathVista, to better highlight its strengths.
>
> A2: Thanks for the suggestion. First, it is as expected that the performance of different methods will become similar with more vision tokens, since the performance will converge to the same position with all vision tokens, i.e., 576. Second, our method shows good potential for low token budget scenarios as in Table 5 of the original submission. Finally, we follow the suggestions and run the evaluation on MMStar as follows. The major issue on these challenging tasks is that the original performance without selection is already quite low. Therefore, it is hard to show the significant difference when the upper-bound is limited. Nevertheless, our method is still better than baselines with a clear margin and is able to recover the baseline with 32 tokens. The discussion can be found in B.3 in revision. For MathVista, the baseline is even lower, which is only 22.6\%. So we did not make a comparison on it.
>
> Table: Comparison on MMStar with LLaVA-1.5-7B.
>
> | Method | Coarse | Fine-Grained | Instance | Logical | Math | Sci&Tech | Average |
> |--------|--------|--------------|----------|---------|------|----------|---------|
> | *Baseline* | | | | | | | |
> | LLaVA-1.5-7B | 63.63 | 25.63 | 38.89 | 28.92 | 26.60 | 18.48 | 33.69 |
> | *64 Tokens* | | | | | | | |
> | VisionZip | 55.27 | 22.92 | 39.32 | 27.95 | 24.76 | 24.62 | 32.47 |
> | DivPrune | 56.35 | 19.50 | 36.72 | 27.73 | 26.70 | 18.94 | 30.99 |
> | Ours | 59.08 | 22.66 | 39.51 | 29.66 | 28.39 | 20.66 | 33.33 |
> | *32 Tokens* | | | | | | | |
> | VisionZip | 48.58 | 19.04 | 39.73 | 29.69 | 22.91 | 21.95 | 30.32 |
> | DivPrune | 54.82 | 21.07 | 37.03 | 27.82 | 24.18 | 19.32 | 30.71 |
> | Ours | 59.56 | 25.71 | 40.49 | 29.94 | 27.92 | 17.37 | 33.50 |
> | *16 Tokens* | | | | | | | |
> | VisionZip | 43.76 | 21.34 | 32.58 | 25.97 | 23.18 | 19.96 | 27.80 |
> | DivPrune | 49.99 | 21.45 | 38.37 | 28.45 | 21.54 | 18.58 | 29.73 |
> | Ours | 56.32 | 21.58 | 39.48 | 30.22 | 23.98 | 15.16 | 31.12 |
> | *8 Tokens* | | | | | | | |
> | VisionZip | 27.93 | 21.26 | 25.71 | 21.84 | 20.18 | 17.83 | 22.46 |
> | DivPrune | 47.25 | 21.05 | 33.89 | 25.75 | 20.32 | 16.76 | 27.50 |
> | Ours | 54.11 | 21.26 | 35.09 | 29.56 | 20.34 | 15.04 | 29.23 |
> | *0 Tokens* | | | | | | | |
> | Baseline | 31.85 | 19.10 | 23.77 | 23.61 | 14.28 | 16.11 | 21.45 |
>
> Q3: Is there a fundamental difference between Coverage Maximization and maximizing Conditional Diversity for Token Pruning[1]?
>
> A3: Thanks for pointing this out. Please kindly note that [1] is a contemporaneous work according to the guidance of ICLR and we are not required to compare with [1]. However, we would like to emphasize that our coverage-based strategy is significantly different from diversity-based methods. First, diversity-based methods focus more on the diversity within the selected subset (i.e., intra-set diversity), while our method is to cover the original set that optimizes the similarity between selected set and the whole set (i.e., inter-set similarity). Compared to our work, [1] is more similar to DivPrune [CVPR'25]. Second, our coverage strategy can bridge multi-modality smoothly by text-vision coverage and vision-vision coverage, where [1] has to encode the text information as a weight for vision tokens, which is not straightforward. If it is necessary, we will add this discussion in the revision.

---

### Official Review · Reviewer_FeoR · 2025-10-28

**Soundness:** 2
**Presentation:** 3
**Contribution:** 2
**Rating:** 4
**Confidence:** 4

**Summary:**

This paper proposes MMTok, a training-free approach for vision token compression in VLMs. They formulate the multimodal token selection problem as a maximum coverage problem, define the coverage and select a subset of vision tokens to cover text tokens and original vision tokens via a greedy approach. Experimentally, they found MMTok achieve better performance compared to other visual token compression approaches on the basis of the LLaVA-Next and Qwen2.5-VL baseline.

**Strengths:**

1. **Reasonable idea.** Using cross-modal correlation for visual token compression is reasonable.

2. **Solid experiments.** MMTok works with LLava-1.5, LLava-Next and Qwen2.5-VL baseline, especially with cutting-edge SOTA VLMs like Qwen2.5-VL. This demonstrates the robustness and effectiveness of this method.

3. **Training-free.** MMTok is training-free and plug-and-play, easy to apply to multiple foundation VLMs.

**Weaknesses:**

1. **Compare with resize baseline.** Experiment results in [1] show that simply resizing the raw image yields good performance and low latency. I am looking forward to the author comparing your approach with the simple resize approach and adding a discussion in the paper.

2. **Work on multi-turn conversation.** MMTok relies on text instruction to prune vision tokens, therefore hard to apply with multi-turn conversation. The author should discuss this situation and propose some solutions.

3. **About the MMTok-Agent.** As shown in Table 2, looks like the performance gain of MMTok-Agent is limited over MMTok, and tha author didn't show this result over Qwen2.5-VL in Table 3.

4. **Runtime.** The author should show the runtime and flops for MMTok.

[1] VisionThink: Smart and Efficient Vision Language Model via Reinforcement Learning

**Questions:**

1. **Performance over reasoning VLMs.** Reasoning VLMs also face the problem of token redundancy. I am curious if MMTok work with reasoning VLMs?

---

> ### Author Response · Authors · 2025-11-26
>
> We would like to thank the reviewer for valuable time and insightful comments. In the uploaded revision, we addressed all raised concerns as follows.
>
> Q1: I am looking forward to the author comparing your approach with the simple resize approach and adding a discussion in the paper.
>
> A1: Thank you for the suggestion, which is a great point. We compare the resize strategy with token budgets as follows. It shows that selecting vision tokens from the original large image can be more effective than resizing under the same token budget. This is because resizing would ignore the redundancy between vision tokens. More importantly, we observe that incorporating with resizing, MMTok can work better than the counter part with the same token budget on full images. For example, with about 10\% token budgets, MMTok with resizing can achieve 2170 on MME while that on original image is only 2051. Exploring resizing with token selection sufficiently can be our future work. We have added this discussion in the revision's appendix B.4.
>
> Table: Comparison with resize strategy on MME with Qwen2.5-VL-7B. The original image is denoted as resize ratio of 1. Qwen has a default minimal number of vision tokens as 256 to obtain meaningful results.
>
> | Model | Image Resize Ratio | Token Avg. | MME P+C ↑ |
> |-------|-------------------|------------|-----------|
> | Qwen2.5-VL-7B | 1 | 867.6 | 2327 |
> | Ours | 1 | 86.8 | 2051 |
> | Ours | 1 | 173.5 | 2217 |
> | *Resize image to fixed ratio of original height and width, respectively* | | | |
> | Qwen2.5-VL-7B | 1/2 | 459.1 | 2274 |
> | Qwen2.5-VL-7B | 1/4 | 349.6 | 2238 |
> | Qwen2.5-VL-7B | 1/8 | 276.0 | 1793 |
> | *Retain 55% Tokens on 1/4 Resized Image* | | | |
> | Ours | 1/4 | 192.3 | 2254 |
> | *Retain 40% Tokens on 1/4 Resized Image* | | | |
> | Ours | 1/4 | 139.8 | 2215 |
> | *Retain 25% Tokens on 1/4 Resized Image* | | | |
> | Ours | 1/4 | 87.4 | 2170 |
>
> Q2: Work on multi-turn conversation. MMTok relies on text instruction to prune vision tokens, therefore hard to apply with multi-turn conversation. The author should discuss this situation and propose some solutions.
>
> A2: Thanks for your suggestion. Our method can handle this scenario benefiting from the multi-modal coverage strategy. In addition to text instruction, we also cover the main content of the original image by vision-vision coverage. Therefore, the selected vision tokens can be reused for different questions. In the revision, we added an example about multi-turn conversation in Figure 3, in which we select tokens at the first turn for Q1, and then reuse the vision tokens in the remaining turns. We can observe that MMTok selects tokens with text from Q1 but vision-vision coverage helps the following questions to be answered correctly with the selected vision tokens.
>
>
> Q3: About the MMTok-Agent. As shown in Table 2, looks like the performance gain of MMTok-Agent is limited over MMTok, and tha author didn't show this result over Qwen2.5-VL in Table 3.
>
> A3: Thanks for pointing this out. Since the main contribution of this work is the maximum coverage for token selection while the agent is optional, we have removed this option in the revision and leave the study of agent for vision token selection as our future work to avoid any misleading.
>
>
> Q4: Runtime. The author should show the runtime and flops for MMTok.
>
> A4: Thank you for the suggestion. Here is a detailed comparison with different numbers of input and selected tokens. For 2880 input tokens, the computation only costs about 13.93 GFLOPs. We have added this discussion in the revision's appendix B.6.
>
> Table: Running time of MMTok. Running time (ms) of MMTok with different numbers of input and selected vision tokens on LLaVA-NeXT-7B. The reported result is averaged over 100 runs on a A6000 GPU.
>
> | #Input | #Select | Time(ms) | #Input | #Select | Time(ms) | #Input | #Select | Time(ms) |
> |--------|---------|----------|--------|---------|----------|--------|---------|----------|
> | 2880 | 160 | 6.417 | 1728 | 96 | 3.862 | 576 | 32 | 1.267 |
> | 2880 | 80 | 3.733 | 1728 | 48 | 2.247 | 576 | 16 | 0.774 |

---

> ### Author Response · Authors · 2025-11-26
>
> Q5: Performance over reasoning VLMs. I am curious if MMTok work with reasoning VLMs?
>
> A5: Thanks for this great suggestion. We conduct the experiments on MMStar dataset as follows. We can observe that vision token selection can also help reasoning tasks. By selecting 32 out of 576 tokens, MMTok is able to recover the performance of the baseline. We have added the result in the revision's appendix B.3.
>
> Table: Comparison on MMStar with LLaVA-1.5-7B
>
> | Method | Coarse | Fine-Grained | Instance | Logical | Math | Sci&Tech | Average |
> |--------|--------|--------------|----------|---------|------|----------|---------|
> | *Baseline* | | | | | | | |
> | LLaVA-1.5-7B | 63.63 | 25.63 | 38.89 | 28.92 | 26.60 | 18.48 | 33.69 |
> | *64 Tokens* | | | | | | | |
> | VisionZip | 55.27 | 22.92 | 39.32 | 27.95 | 24.76 | 24.62 | 32.47 |
> | DivPrune | 56.35 | 19.50 | 36.72 | 27.73 | 26.70 | 18.94 | 30.99 |
> | Ours | 59.08 | 22.66 | 39.51 | 29.66 | 28.39 | 20.66 | 33.33 |
> | *32 Tokens* | | | | | | | |
> | VisionZip | 48.58 | 19.04 | 39.73 | 29.69 | 22.91 | 21.95 | 30.32 |
> | DivPrune | 54.82 | 21.07 | 37.03 | 27.82 | 24.18 | 19.32 | 30.71 |
> | Ours | 59.56 | 25.71 | 40.49 | 29.94 | 27.92 | 17.37 | 33.50 |
> | *16 Tokens* | | | | | | | |
> | VisionZip | 43.76 | 21.34 | 32.58 | 25.97 | 23.18 | 19.96 | 27.80 |
> | DivPrune | 49.99 | 21.45 | 38.37 | 28.45 | 21.54 | 18.58 | 29.73 |
> | Ours | 56.32 | 21.58 | 39.48 | 30.22 | 23.98 | 15.16 | 31.12 |
> | *8 Tokens* | | | | | | | |
> | VisionZip | 27.93 | 21.26 | 25.71 | 21.84 | 20.18 | 17.83 | 22.46 |
> | DivPrune | 47.25 | 21.05 | 33.89 | 25.75 | 20.32 | 16.76 | 27.50 |
> | Ours | 54.11 | 21.26 | 35.09 | 29.56 | 20.34 | 15.04 | 29.23 |
> | *0 Tokens* | | | | | | | |
> | Baseline | 31.85 | 19.10 | 23.77 | 23.61 | 14.28 | 16.11 | 21.45 |

---

> ### Comment · Reviewer_FeoR · 2025-11-26
> **Response to Author Rebuttal**
>
> Thanks for the author's response.
> The author addressed most of my concerns, on image resizing, multi-turn conversation, and running time.
>
> The discussion and experiment results over image resizing are insightful and interesting. Specifically, the author achieves a better performance when applying MMTok on a resized image with a higher token keep rate. These results will bring more insights to the community.
>
> Overall, I will raise my score to 6.

---

> > ### Author Response · Authors · 2025-11-26
> >
> > We highly appreciate your prompt response and are very glad that most of your concerns have been addressed. Your suggestions on image resizing and multi-turn conversation are very insightful, which help improve this work significantly. Thanks for your encouraging raise of the score.

---

### Official Review · Reviewer_8R44 · 2025-11-01

**Soundness:** 2
**Presentation:** 3
**Contribution:** 2
**Rating:** 4
**Confidence:** 4

**Summary:**

This paper proposes MMTok, a training-free method for vision token selection in Vision-Language Models (VLMs) by formulating the problem as a multimodal maximum coverage problem.  The approach leverages both text and vision tokens to select a subset of vision tokens that maximally cover semantic information from both modalities.  The method is evaluated across multiple VLMs and benchmarks, showing strong performance in maintaining accuracy while reducing token counts and improving inference speed.

**Strengths:**

-	The paper addresses an important and practical problem: improving inference efficiency of VLMs without fine-tuning. And the multimodal coverage formulation is well-motivated, combining both text-vision and vision-vision similarity in a principled manner.

-	Extensive experiments across multiple models (LLaVA-1.5, LLaVA-NeXT, Qwen) and datasets demonstrate the generality and effectiveness of the method.

-	The method achieves impressive results, e.g., 1.87× speedup on LLaVA-NeXT-13B with 98.7% performance retention, and maintains strong performance even with very few tokens (e.g., 4 tokens on LLaVA-1.5-7B).

**Weaknesses:**

-	My main concern lies in the efficiency part. The authors only provide efficiency test on base version. The agent-based text enrichment is presented as optional and shows mixed results, but its computational overhead and when it is most beneficial are not thoroughly discussed.
-	What is the efficiency cost under the practical experimental settings? e.g. The experiments in Table 3 with Qwen2.5-VL.
-	How does the time complexity of MMTok scale with the number of vision tokens, especially compared to the baselines?
-	The visualization results are not fully convincing, which weaken the interpretability of the proposed approach.
-	The performance drop is severe under some benchmarks, which yield questions about the availability among real-world scenario.

**Questions:**

See the weaknesses part.

---

> ### Author Response · Authors · 2025-11-26
>
> We would like to thank the reviewer for valuable time and insightful comments. In the uploaded revision, we addressed all raised concerns as follows.
>
> Q1: The agent-based text enrichment is presented as optional and shows mixed results, but its computational overhead and when it is most beneficial are not thoroughly discussed.
>
> A1: Thank you for the suggestion. Since the main contribution of this work is the maximum coverage for token selection while the agent is optional, we focus on the efficiency of the major proposal without agents in the submission. To avoid any misleading, we removed the agent version in the revision and leave the study of agent for token selection as our future work.
>
> Q2: What is the efficiency cost under the practical experimental settings? e.g. The experiments in Table 3 with Qwen2.5-VL.
>
> A2: Following the common practice as VisionZip, we conducted the efficiency comparison under practical experimental settings in Table 7 of the original submission. The running time is reported from the real data sets with end-to-end inference. For curiosity on Qwen2.5-VL-7B, in the revision's appendix B.5, we add the results of Qwen2.5-VL-7B as follows. We find that vision token selection can also accelerate the inference of state-of-the-art VLMs. For the running time of MMTok itself, we will discuss in the answer to the next question.
>
> Table: Comparison of Inference Efficiency on Qwen2.5-VL-7B (The initial memory usage for loading the model is 15.87GB.)
>
> | Model | Token Avg. | Inference Time(s) | GPU util. | Memory (+15.87GB) |
> |-------|------------|-------------------|-----------|-------------------|
> | *1 × A6000 GPU Performance on MME* | | | | |
> | Qwen2.5-VL-7B | 867.6 | 675 | 77.0% | 3.05 |
> | VisionZip | 86.8 | 508 | 66.3% | 0.41 |
> | DivPrune | 86.8 | 423 | 55.1% | 0.71 |
> | Ours | 86.8 | 419 | 60.0% | 0.71 |
>
> Q3: How does the time complexity of MMTok scale with the number of vision tokens, especially compared to the baselines?
>
> A3: In MMTok, the similarity matrix can be constructed efficiently using the pytorch built-in libraries. Then, for token selection, MMTok proposes a greedy algorithm only with max operations and can run in $O(kn)$ to pick $k$ vision tokens, where $n$ is the total number of vision tokens. Therefore, our method can scale well with vision tokens and with the number of selected vision tokens. Below we show the running time of MMTok with the varying number of input and selected vision tokens. We find that with the same number of input tokens, the running time is almost linear in the number of selected tokens, which confirms our analysis. Moreover, even with 2880 input tokens, the running time of MMTok is less than 7ms, which is negligible for real applications. Thanks for your suggestion, we have added this discussion in the revision's appendix B.6.
>
> Table: Running time of MMTok (Running time (ms) of MMTok with different numbers of input and selected vision tokens on LLaVA-NeXT-7B. The reported result is averaged over 100 runs on a A6000 GPU.)
>
> | #Input | #Select | Time(ms) | #Input | #Select | Time(ms) | #Input | #Select | Time(ms) |
> |--------|---------|----------|--------|---------|----------|--------|---------|----------|
> | 2880 | 160 | 6.417 | 1728 | 96 | 3.862 | 576 | 32 | 1.267 |
> | 2880 | 80 | 3.733 | 1728 | 48 | 2.247 | 576 | 16 | 0.774 |
>
> Q4: The visualization results are not fully convincing, which weaken the interpretability of the proposed approach.
>
> A4: In Fig. 3 of the original submission (now Fig. 5 in the revision), we aim to visualize the top selected patches by MMTok and those of DivPrune. From the columns (b) and (c), we can observe that MMTok selects top patches according to the word to patch similarity, which is aligning well with the question semantically. In contrast, as shown in columns (d) and (e), DivPrune selected top patches without any close semantic relation to the question. This further demonstrates that MMTok can help significantly reduce the number of tokens without losing the semantic relation to the questions, so as to provide better performance. Thanks for pointing this out, we have updated the discussion in the revision for clarification.
>
> Q5: The performance drop is severe under some benchmarks, which yield questions about the availability among real-world scenario.
>
> A5: Thanks for pointing this out. Please kindly note that the performance drop mainly depends on how many vision tokens are retained. According to Table 1 of the original submission, about 96.6\% performance can be kept on average when using 64 out of 576 tokens. The performance loss becomes significant when the token budget is reduced further as in Table 5, which is as expected. Moreover, according to Table 7, even with up to 160 out of 2880 tokens selected, MMTok can still provide efficiency benefits for real-world tasks. If we did not accurately capture your concern, please kindly help specify.

---

### Author Response · Authors · 2025-12-03

Dear ACs:

We sincerely thank all reviewers again for their valuable time and thoughtful feedback, which has helped improve the uploaded revision. In summary, in the revised version, we have made the following changes to address all concerns raised:

1)	Concerns about the MMTok-Agent (raised by reviewer 8R44, FeoR, TSCU, and EEd7): Since the main contribution of this work is the maximum coverage for token selection and the agent is optional, we removed this option in the revision to avoid any misleading.

2)	Efficiency cost of Qwen2.5-VL (raised by reviewer 8R44): We added the efficiency results of Qwen2.5-VL-7B in B.5 of the revision, which shows that our proposal can also help accelerate the inference of state-of-the-art VLMs.

3)	Time complexity or runtime with the number of vision tokens (raised by reviewer 8R44 and FoeR): In the revision’s B.6, we added the discussion of the time complexity and the experiments with varying number of input and selected vision tokens. We find that the running time is almost linear in the number of selected tokens and even with 2880 input tokens, the running time of our proposal is less than 7ms.

4)	Comparison to simple resize approach (raised by reviewer FoeR): In revision’s B.4, we compare with the resize approach. We find that vision token selection is more effective than resizing under the same token budget. More importantly, incorporating our proposal with resizing can further improve the performance.

5)	Evaluation on multi-turn conversation (raised by reviewer FoeR): In revision’s experiment section, we added this discussion with an example about multi-turn conversation in Figure 3, in which we select vision tokens only at the first turn and then reuse in the remaining turns, which can still help all questions answered correctly.

6)	Evaluation on harder tasks such as reasoning VLMs (raised by reviewer FoeR and TSCU): In revision’s B.3, we added experiments on reasoning datasets such as MMStar, in which we can observe that our proposal can also help reasoning tasks.

7)	Applying the proposal to decoder (raised by reviewer EEd7): In revision’s B.7, we added the experiments on decoder. We find that applying our proposal to decoder can keep the similar performance while further improving efficiency.

8)	Individual case illustration (raised by reviewer EEd7): We added some examples in Figure 4 of the revision and find that the answer changes when the number of vision tokens is reduced significantly. Moreover, for critical or harder questions, the experiments suggest keeping more vision tokens.

During the discussion period, reviewer 8R44 and TSCU did not have a chance to reply to our rebuttal unfortunately. Reviewer FoeR and EEd7 replied that their concerns were effectively resolved. Moreover, reviewer FoeR commented that **''the discussion and experiment result over image resizing are insightful and interesting, which can bring more insights to the community''.** Therefore, reviewer FoeR raised the score from **4 to 6**. Finally, it can be noted that although reviewer 8R44 and TSCU did not have a chance to reply, their major concerns are **shared** with the other two reviewers, who replied that concerns were effectively **resolved**.

Sincerely,

The Authors

---

### Meta-Review · Area_Chair_E1Jh · 2025-12-03

**Summary:**

This paper proposes MMTok, a training-free method to improve Vision-Language Model (VLM) inference efficiency by selecting a subset of informative vision tokens. It formulates token selection as a multimodal maximum coverage problem, optimizing selected tokens to cover both the text tokens and the original vision tokens. Evaluations across models like LLaVA and Qwen2.5-VL show significant speedups (e.g., 1.87×) with minimal performance loss. Reviewers initially raised several concerns. These included questions about the optional MMTok-Agent's utility and overhead, the method's efficiency cost on state-of-the-art VLMs like Qwen2.5-VL, its time complexity, the lack of comparison with a simple image resizing baseline, applicability to multi-turn conversations and reasoning tasks, extension to the decoder, unconvincing visualizations, and analysis of individual case performance. The authors' rebuttal and revised manuscript addressed most points. They removed the MMTok-Agent to focus on the core contribution. They added efficiency results for Qwen2.5-VL, a runtime analysis showing linear scaling and low overhead (~7ms), and a comparison with resizing, finding token selection more effective. They discussed multi-turn conversation handling, added experiments on the reasoning dataset MMStar, provided preliminary results for decoder application, and included case studies showing answer changes under aggressive pruning. Reviewers FeoR and EEd7 found their concerns largely resolved, with FeoR increasing the score from 4 to 6.

**Reviewer Concerns:**

Most reviewer concerns, including those about efficiency analysis, comparison to resizing, multi-turn conversations, reasoning tasks, and the optional agent, were directly addressed in the revision. The remaining minor points, such as further exploration on very challenging benchmarks, do not undermine the paper's core contributions.

**Reviewer Scores:**

Reviewer FeoR raised the score from 4 to 6 after the rebuttal. Reviewer EEd7 maintained a score of 6. While Reviewers 8R44 and TSCU did not finalize their post-rebuttal scores, their primary concerns overlapped with those addressed by the other reviewers, making a positive score adjustment likely had they participated fully.

---

### Decision · Program_Chairs · 2026-01-26

Accept (Poster)